# ENHANCING GROUP FAIRNESS IN ONLINE SETTINGS USING OBLIQUE DECISION FORESTS

**Somnath Basu Roy Chowdhury**[*,1]**, Nicholas Monath**[2]**, Ahmad Beirami**[3]**, Rahul Kidambi**[3]**, Avinava Dubey**[3]**, Amr Ahmed**[3]**, Snigdha Chaturvedi**[1]

[1]UNC Chapel Hill, [2]Google DeepMind, [3]Google Research.
`{somnath, snigdha}@cs.unc.edu`
`{nmonath, beirami, rahulkidambi, avinavadubey, amra}@google.com`

## ABSTRACT

Fairness, especially group fairness, is an important consideration in the context of machine learning systems. The most commonly adopted group fairness-enhancing techniques are in-processing methods that rely on a mixture of a fairness objective (e.g., demographic parity) and a task-specific objective (e.g., cross-entropy) during the training process. However, when data arrives in an online fashion – one instance at a time – optimizing such fairness objectives poses several challenges. In particular, group fairness objectives are defined using expectations of predictions across different demographic groups. In the online setting, where the algorithm has access to a single instance at a time, estimating the group fairness objective requires additional storage and significantly more computation (e.g., forward/backward passes) than the task-specific objective at every time step. In this paper, we propose *Aranyani*, an ensemble of oblique decision trees, to make fair decisions in online settings. The hierarchical tree structure of *Aranyani* enables parameter isolation and allows us to efficiently compute the fairness gradients using aggregate statistics of previous decisions, eliminating the need for additional storage and forward/backward passes. We also present an efficient framework to train *Aranyani* and theoretically analyze several of its properties. We conduct empirical evaluations on 5 publicly available benchmarks (including vision and language datasets) to show that *Aranyani* achieves a better accuracy-fairness trade-off compared to baseline approaches.

## 1 INTRODUCTION

Critical applications of machine learning, such as hiring (Dastin, 2022) and criminal recidivism (Larson et al., 2016), require special attention to avoid perpetuating biases present in training data (Corbett-Davies et al., 2017; Buolamwini & Gebru, 2018; Raji & Buolamwini, 2019). Group fairness, which is a well-studied paradigm for mitigating such biases in machine learning (Mehrabi et al., 2021; Hort et al., 2023), tries to achieve statistical parity of a system's predictions among different demographic (or protected) groups (e.g., gender or race). In general, group fairness-enhancing techniques can be broadly categorized into three categories: pre-processing (Zemel et al., 2013; Calmon et al., 2017), post-processing (Hardt et al., 2016; Pleiss et al., 2017; Alghamdi et al., 2022), and in-processing (Quadrianto & Sharmanska, 2017; Agarwal et al., 2018; Lowy et al., 2022; Baharlouei et al., 2024) techniques. Most of these approaches rely on group fairness objectives that are optimized alongside task-specific objectives in an offline setting (Dwork et al., 2012). Group fairness objectives (e.g., demographic parity) are defined using expectations of predictions across different demographic groups, requiring access to labeled data from different groups. However, in many modern applications (e.g., output moderation using toxicity classifiers for chatbots, social media content, etc.), data arrives in an online fashion. In such cases, the definition of safety is evolving, and new unsafe data points are identified on the fly, making them prime candidates for online learning.

In the online setting, optimizing for group fairness poses several unique challenges. Central to this paper, in-processing techniques require additional storage or computation since the system only has

---

*Work done during an internship at Google Research.

access to a single input instance at any given time. In online settings, naively training the model using group fairness loss involves storing all (or at least a subset of) the input instances seen so far, and performing forward/backward passes through the model using these instances at each step of online learning, which can be computationally expensive. We also note that other techniques such as pre-processing techniques are clearly impractical as they require prior access to data. Post-processing techniques typically assume black-box access to a trained model and a held-out validation set (Hardt et al., 2016), which can be impractical or expensive to acquire during online learning.

In this paper, we present a novel framework *Aranyani*, which consists of an ensemble of oblique decision trees. *Aranyani* uses the structural properties of the tree to enhance group fairness in decisions made during online learning. The final prediction in oblique trees is a combination of local decisions at individual tree nodes. We show that imposing group fairness constraints on local node-level decisions results in parameter isolation, which empirically leads to better and fairer solutions. In the online setting, we show that maintaining the aggregate statistics of the local node-level decisions allows us to efficiently estimate group fairness gradients, eliminating the need for additional storage or forward/backward passes. We present an efficient framework to train *Aranyani* using state-of-the-art autograd libraries and modern accelerators. We also theoretically study several properties of *Aranyani* including the convergence of gradient estimates. Empirically, we observe that *Aranyani* achieves the best accuracy-fairness trade-off on 5 different online learning setups.

Our paper is organized as follows: (a) We begin by introducing the fundamentals of oblique decision trees and provide the details of oblique decision forests used in *Aranyani* (Section 2), (b) We describe the problem setup and discuss how to enforce group fairness in the simpler offline setting (Section 3.1 & 3.2), (c) We describe the functioning of *Aranyani* in the online setting (Section 3.3), (d) We describe an efficient training procedure for oblique decision forests that enables gradient computation using back-propagation (Section 3.4), (e) We theoretically analyze several properties of *Aranyani* (Section 4), and (f) We describe the experimental setup and results of *Aranyani* and other baseline approaches on several datasets (Section 5). We observe that *Aranyani* achieves the best accuracy-fairness tradeoff, and provides significant time and memory complexity gains compared to naively storing input instances to compute the group fairness loss.

## 2 OBLIQUE DECISION TREES

We introduce our proposed framework, *Aranyani*, an ensemble of oblique decision trees, for achieving group fairness in an online setting. In this section, we introduce the fundamentals of oblique decision trees and discuss the details of the prediction function used in *Aranyani*. Similar to a conventional decision tree, an oblique decision tree splits the input space to make predictions by routing samples through different paths along the tree. However, unlike a decision tree, which only makes axis-aligned splits, an oblique decision tree can make arbitrary oblique splits by using routing functions that consider all input features. The routing functions in oblique decision tree nodes can be parameterized using neural networks (Murthy et al., 1994; Jordan & Jacobs, 1994). This allows it to potentially fit arbitrary boundary structures more effectively. We formally describe the details of the oblique decision tree structure below:

**Definition 1** (Oblique binary decision tree (Karthikeyan et al., 2022)). *An oblique tree of height $h$ represents a function $f(\mathbf{x}; \mathbf{W}, \mathbf{B}, \mathbf{\Theta}) : \mathbb{R}^d \to \mathbb{R}^c$ parameterized by $\mathbf{w}_{ij} \in \mathbb{R}^d$, $\mathbf{b}_{ij} \in \mathbb{R}$ at $(i, j)$-th node ($j$-th node at depth $i$). Each node computes $n_{ij}(\mathbf{x}) = \mathbf{w}_{ij}^T \mathbf{x} + \mathbf{b}_{ij} > 0$, which decides whether $\mathbf{x}$ must traverse the left or right child. After traversing the tree, input $\mathbf{x}$ arrives at the $l$-th leaf that outputs $\theta_l \in \mathbb{R}^c$ ($c > 1$ for classification and $c = 1$ for regression).*

The oblique tree parameters $(\mathbf{W}, \mathbf{B}, \mathbf{\Theta})$ can be learned using gradient descent (Karthikeyan et al., 2022). However, the hard routing in oblique decision trees ($\mathbf{x}$ is routed either to the left or right child) makes the learning process non-trivial. In our work, we consider a modified soft version of oblique trees where an input $\mathbf{x}$ is routed to both left and right child at every tree node with certain probabilities based on the node output, $n_{ij}(\mathbf{x})$.

**Definition 2** (Soft-Routed Oblique binary decision tree). *Using the same parameterization in Definition 1, soft-routed oblique trees route $\mathbf{x}$ to both children at each node with a certain probability. At $(i, j)$-th node, the probability that $\mathbf{x}$ is routed to the left node $p(\swarrow) = n_{ij}(\mathbf{x})$, and the right node is $p(\searrow) = 1 - n_{ij}(\mathbf{x})$, where $n_{ij}(\mathbf{x}) = g(\mathbf{w}_{ij}^T \mathbf{x} + \mathbf{b}_{ij})$ and $g(x) \in [0, 1]$ is an activation function. The output $f(\mathbf{x}) = \sum_l p_l(\mathbf{x})\theta_l$, where $p_l(\mathbf{x})$ is the probability with which $\mathbf{x}$ reaches the $l$-th leaf.*

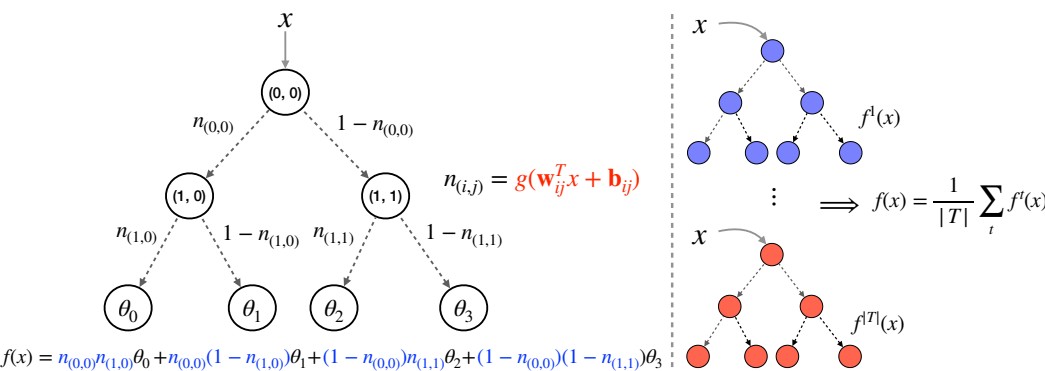

Figure 1: Schematic diagram of the functioning of Oblique Decision Forests. (Left): We illustrate the computation of a soft-routed oblique tree output $f^t(\mathbf{x})$ using individual tree node outputs. We observe that the final tree decision is composed of individual node outputs. (Right): We showcase how decisions from multiple oblique trees are combined to form $f(\mathbf{x})$.

In soft-routed oblique decision trees, we observe that the prediction $f(\mathbf{x})$ is a linear combination of all the leaf parameters. The coefficients $p_l(\mathbf{x}) = \prod_{i=1}^{h} n_{i,A(i,l)}(\mathbf{x})$ is the product of all probabilities along the path from the root to the $l$-th leaf and $A(i,l)$ is the $l$-th leaf's ancestor at depth $i$. We observe that learning the parameters of the soft-routed tree structure is much easier as we can easily compute the gradients of parameters w.r.t. $f(\mathbf{x})$ using backpropagation. We further improve the efficiency by computing $f(\mathbf{x})$ using matrix operations as described in Section 3.4. In our work, we use a complete binary tree of height $h$ to parameterize obliques trees. Note that the proposed soft-routed oblique trees are reminiscent of the sigmoid tree decomposition (Yang et al., 2019) used in alleviating the softmax bottleneck.

We use an ensemble of trees and the expected output as the final prediction to reduce the variance and increase the predictive power of the outputs of soft-routed oblique decision trees. We call this *soft-routed oblique forest*, which is computed as: $f(\mathbf{x}) = 1/|\mathcal{T}| \sum_{t=1}^{|\mathcal{T}|} f^t(\mathbf{x})$, where $f^t(\mathbf{x})$ is the output of the $t$-th soft-routed oblique decision tree. The schematic diagram is shown in Figure 1.

## 3    ARANYANI: FAIR OBLIQUE DECISION FORESTS

In this section, we present *Aranyani*, a framework to enhance group fairness in decisions made during online learning. In this work, we focus on the group fairness notion of statistical or demographic parity. Our framework can be easily extended to other notions of group fairness, such as equalized odds (Hardt et al., 2016), equal opportunity (Hardt et al., 2016), and representation parity (Hashimoto et al., 2018) as described in Appendix C.1.

### 3.1    PROBLEM FORMULATION

We describe the online learning setup where input instances arrive incrementally $\{\mathbf{x}_1, \mathbf{x}_2, \ldots\}$, with $\mathbf{x}_t$ arriving at time step $t$. The goal of the oblique decision forest $f$ is to make accurate decisions w.r.t. the *task*, $y$ (e.g., hiring decisions) while being fair w.r.t. the *protected attribute*, $a$ (e.g., gender). At time step $t$, $f$ outputs a prediction $\hat{y}_t$ based on $f(\mathbf{x}_t)$, where $\hat{y}_t = f(\mathbf{x}_t)$ for regression and $\hat{y}_t = \arg\max f(\mathbf{x})$ for classification. With a slight abuse of notation, we denote decisions made for an instance $\mathbf{x}$ with protected attribute $a = k$ as $f(\mathbf{x}|a = k)$ (forest output) or $n(\mathbf{x}|a = k)$ (node output), where $k = \{0, 1\}$. Following prior work (Zhang & Ntoutsi, 2019), we consider the setup where the model receives both the true label $y_t$ and demographic label $a_t$ of an instance $\mathbf{x}_t$ after predicting $\hat{y}_t$. The model can then use this feedback to update its parameters. In this work, we consider the scenario where the model is not allowed to store previous samples. Note that storing previous instances may pose additional challenges in applications that need to adhere to privacy guidelines (Voigt & Von dem Bussche, 2017) or involve distributed infrastructure, such as federated learning (Konečný et al., 2016). In this work, we focus on demographic parity notion of fairness:

$$\mathrm{DP} = |\mathbb{E}[f(\mathbf{x}|a = 0)] - \mathbb{E}[f(\mathbf{x}|a = 1)]|. \tag{1}$$

Note that in the above definition, we consider a slightly modified version of demographic parity to handle scenarios where the preferred outcome (or target label) is not explicitly defined. For simplicity, we describe our system using a binary protected attribute $a \in \{0, 1\}$, however, it can handle protected attributes with multiple classes ($>2$) as well (see more details in Appendix C.2).

## 3.2 OFFLINE SETTING

We begin by describing the simpler offline setting, in which all the training data is accessible prior to making predictions. In this setting, $f$ is optimized using stochastic gradient descent in a batch-wise manner. The constrained objective function is shown below:

$$\min_f \mathcal{L}(f(\mathbf{x}), y), \text{ subject to } |\mathbb{E}[f(\mathbf{x}|a = 0)] - \mathbb{E}[f(\mathbf{x}|a = 1)]| < \epsilon, \tag{2}$$

where $\mathcal{L}(\cdot, \cdot)$ is the task loss function[1] (e.g., cross entropy loss) and $y$ is the true task label. The non-convex and non-smooth nature of the fairness objective (L1-norm in the DP term) makes it difficult to optimize the group fairness loss.

When $f$ is an oblique decision forest, we leverage its hierarchical prediction structure to impose group fairness constraints on the local node-level outputs, $n_{ij}(\mathbf{x})$ ($j$-th node at depth $i$) of the tree. The rationale behind applying constraints at the node outputs stems from the observation that if instances from different groups receive similar decisions at every node, then the final decision (which is an aggregation of the local decisions, Definition 2) is expected to be similar. We can formulate the node-level fairness constraints ($\mathcal{F}_{ij}$) as shown below:

$$\min_f \mathcal{L}(f(\mathbf{x}), y), \text{ s.t. } \forall (i, j) |\mathcal{F}_{ij}| < \epsilon, \text{ where } \mathcal{F}_{ij} = \mathbb{E}[n_{ij}(\mathbf{x}|a = 0)] - \mathbb{E}[n_{ij}(\mathbf{x}|a = 1)]. \tag{3}$$

The node constraints $|\mathcal{F}_{ij}|$ are applied to all nodes (indexed by $(i, j)$) in the tree. We discuss the relation between node-level constraints and group fairness in Section 4. We note that $\mathcal{F}_{ij}$ is a function of the parameters of the $(i, j)$-th node only. In practice, we observe that this parameter isolation (Rusu et al., 2016) achieved by imposing fairness constraints on the local node-level outputs makes it easier to optimize $f$. We would like to emphasize that *Aranyani* can be extended to other notions of group fairness by modifying the formulation of $\mathcal{F}_{ij}$ (see Appendix C.1).

However, the optimization procedure in Equation 3 is hard to solve. We relax it by using a smooth surrogate for the L1-norm and turning the constraint into a regularizer. In particular, we use Huber loss function (Huber, 1992) (with hyperparameter $\delta > 0$) and the relaxed optimization objective is:

$$\min_f \left\{ \mathcal{L}(f(\mathbf{x}), y) + \lambda \sum_{i,j} H_\delta(\mathcal{F}_{ij}) \right\}, \text{ where } H_\delta(\mathcal{F}_{ij}) := \begin{cases} \mathcal{F}_{ij}^2/2, & \text{if } |\mathcal{F}_{ij}| < \delta \\ \delta|\mathcal{F}_{ij} - \delta/2|, & \text{otherwise} \end{cases}, \tag{4}$$

with $\lambda$ being a hyperparameter. In the offline setting, the expectations over input instances in $\mathcal{F}_{ij}$ (Equation 3) are computed using samples within a training batch. In the online setup, computing these expectations is challenging as we only have access to individual instances at a time and not to a batch. Therefore, naively optimizing Equation 3 or 4 in the online setup requires storing all (or at least a subset) of the input instances. Moreover, we need to perform additional forward and backward passes for all stored instances to compute the group fairness gradients. In practice, this can be quite expensive in the online setting. In the following section, we discuss how to efficiently compute these group fairness gradients in the online setting.

## 3.3 ONLINE SETTING

In this section, we describe the training process for *Aranyani* in the online setting. As noted in the previous section, computing the expectations in group fairness terms is challenging due to storage and computational costs. However, we do not need to compute the loss exactly using previous input instances as we only need the gradients of the loss function to update the model. We show that it is possible to estimate the fairness gradients by maintaining aggregate statistics of the node-level outputs in the tree. Taking the derivative of Equation 4, with respect to node parameters $\Theta \in [\mathbf{W}, \mathbf{B}]$

---

[1]We assume that the task loss can be defined using a single instance. This holds for most commonly used loss functions like cross entropy and mean squared error.

of model $f$, we get the following gradients:

$$G(\Theta) = \nabla_\Theta \mathcal{L}(f(\mathbf{x}), y) + \lambda \sum_{\forall i, j} \nabla_\Theta H_\delta(\mathcal{F}_{ij}),$$

$$\text{where } \nabla_\Theta H_\delta(\mathcal{F}_{ij}) = \begin{cases} \mathcal{F}_{ij} \nabla_\Theta \mathcal{F}_{ij}, & \text{if } |\mathcal{F}_{ij}| < \delta \\ \delta \text{sgn}\left(\mathcal{F}_{ij} - \delta/2\right) \nabla_\Theta \mathcal{F}_{ij}, & \text{otherwise} \end{cases} \quad (5)$$

$$\text{and } \nabla_\Theta \mathcal{F}_{ij} = \mathbb{E}[\nabla_\Theta n_{ij}(\mathbf{x}|a=0)] - \mathbb{E}[\nabla_\Theta n_{ij}(\mathbf{x}|a=1)].$$

In the above equation, we have an unbiased estimate of the task gradient: $\nabla_\Theta \mathcal{L}(f(\mathbf{x}), y)$ for an i.i.d. sample $\mathbf{x}$. The fairness gradient $\nabla_\Theta H_\delta(\mathcal{F}_{ij})$ can be estimated by maintaining a number of aggregate statistics at each decision tree node. Specifically, we need to store the following aggregate statistics: (a) $\mathbb{E}[n_{ij}(\mathbf{x}|a=k)]$ and (b) $\mathbb{E}[\nabla_\Theta n_{ij}(\mathbf{x}|a=k)], \forall k$, where $k \in \{0, 1\}$ denotes different protected attribute labels. In practice, for every incoming sample $\mathbf{x}_t$ we compute $n_{ij}(\mathbf{x}_t)$ and $\nabla_\Theta n_{ij}(\mathbf{x}_t)$, and update the aggregate statistics based on the protected label $a_t$. We denote the node constraints and gradients estimated using aggregate statistics as $\widehat{\mathcal{F}}_{ij}$ and $\widehat{G}(\Theta)$ respectively.

Note that in this setup, we do not need to store the previous input instances or query $f$ multiple times. Furthermore, computing $n_{ij}(\mathbf{x})$ and $\nabla_\Theta n_{ij}(\mathbf{x})$ is relatively inexpensive. For sigmoid functions, both $n_{ij}(\mathbf{x})$ and $\nabla_\Theta n_{ij}(\mathbf{x})$ can be obtained using a forward pass as: $\nabla_\Theta n_{ij}(\mathbf{x}) = n_{ij}(\mathbf{x})(1 - n_{ij}(\mathbf{x}))$. We discuss several properties of the estimated gradient $\widehat{G}(\Theta)$ in Section 4.

### 3.4 Training Procedure

In this section, we present an efficient training strategy for *Aranyani*. In general, tree-based architectures are slow to train using gradient descent as gradients need to propagate from leaf nodes to other nodes in a sequential fashion. We introduce a parameterization that enables us to compute oblique tree outputs only using matrix operations. This enables training on modern accelerators (like GPUs or TPUs) and helps us to efficiently compute task gradients (Equation 5) by using state-of-the-art autograd libraries. We begin by noting that all tree node outputs are independent of each other given the input $\mathbf{x}$. Therefore, the node outputs can be computed in parallel as shown below:

$$\mathbf{N} = g(\mathbf{W}^T \mathbf{x} + \mathbf{B}) \in \mathbb{R}^m, \text{ where } \mathbf{W} \in \mathbb{R}^{m \times d}, \mathbf{B} \in \mathbb{R}^m$$

and $m = 2^h - 1$ is the number of internal nodes (for a complete binary tree). Subsequently, these node outputs are utilized to calculate the probabilities required to reach individual leaf nodes. The path probabilities are computed by creating $2^h$ (number of leaf nodes) copies of the node outputs, $\overline{\mathbf{N}} = (\mathbf{N}, \mathbf{N}, \cdots, \mathbf{N}) \in \mathbb{R}^{m \times 2^h}$, and applying a mask, $\mathbf{A}$, that selects the ancestors for each leaf node. Each element of the mask $\mathbf{A}_{ij} \in \{-1, 0, 1\}$ selects whether the leaf path from a selected node is left (1), right (-1), or doesn't exist (0). The exact probabilities are then stored in $\mathbf{P}$. This sequence of operations is shown below:

$$f(\mathbf{x}) = \exp\left(\mathbf{1}_{1 \times m} \log \mathbf{P}\right) \Theta, \text{ where } \mathbf{P} = \text{ReLU}(\overline{\mathbf{N}} \odot \mathbf{A}) + (\mathbf{1}_{m \times 2^h} - \text{ReLU}(-\overline{\mathbf{N}} \odot \mathbf{A}))$$

and $\Theta \in \mathbb{R}^{2^h \times c}$. The selected probabilities can be utilized to compute the final output as $f(\mathbf{x})$. More details about the construction of mask $\mathbf{A}$ is reported in Appendix D.3.

## 4 Theoretical Analysis

In this section, we theoretically analyze several properties of *Aranyani*. In our proofs, we make assumptions that are standard in the optimization literature such as compact parameter set, Lipschitz task loss, bounded input $\mathbf{x}$, and bounded gradient noise (see Appendix A.1 for more details). First, we discuss the conditions of the node-level decisions and how they relate to group fairness constraints. Second, we analyze the properties of the gradient estimates (Equation 5) and show that the expected gradient norm converges for small step size and large enough time steps.

**Lemma 1** (Demographic Parity Bound). *Let $f(\mathbf{x})$ be a soft-routed oblique decision tree of height $h$ with $\|\theta_l\| = 1$ and assume an equal number of input instances $\mathbf{x}$ for each group of a binary protected attribute $a \in \{0, 1\}$. Then, if all the node-level decisions satisfy the following condition:*

$$\mathbb{E}[|n_{ij}(\mathbf{x}|a=0) - n_{ij}(\mathbf{x}|a=1)|] \leq \epsilon, \ \forall(i, j). \quad (6)$$

*Then, the overall demographic parity of $f(\mathbf{x})$ is bounded as: $\text{DP} \leq h2^h \epsilon$, for $\epsilon > 0$.*

The above lemma (proof in Appendix A.2) provides the node-level constraint (Equation 6) that upper bounds the demographic parity. We note that the node constraint $|\mathcal{F}_{ij}|$ (Equation 4) is a weaker constraint than the one derived above. The rationale behind using $\mathcal{F}_{ij}$ over the derived constraint is based on two key considerations: First, the expectation is computed using sample pairs from complementary groups (in Equation 6), which is challenging to compute in both offline and online settings. Second, optimizing this constraint can severely limit the task performance as it encourages the trivial solution of having the same node outputs for all instances.

Next, we derive the Rademacher complexity of soft-routed decision trees. Empirical Rademacher complexity, $\hat{R}_n(\mathcal{H})$, measures the ability of function class $\mathcal{H}$ to fit random noise indicating its expressivity (formal definition and proof in Appendix A.3).

**Lemma 2** (Rademacher Complexity). *Empirical Rademacher complexity, $\hat{R}_n(\mathcal{H})$, for soft-routed decision tree (of height $h$) function class, $f(\mathbf{x}) \in \mathcal{H}$, and $\|\theta_l\| = 1$ is bounded as: $\hat{R}_n(\mathcal{H}) \leq 2^h/\sqrt{n}$.*

We observe that the bound exponentially increases with the height of the tree, $h$. Interestingly, according to the DP bound in Equation 6, it appears that we can easily improve group fairness by using a shallower tree (smaller $h$). This illustrates the trade-off between fairness and accuracy, highlighting that it is not feasible to enhance group fairness without a substantial impact on accuracy.

Next, we derive the estimation error bound for the gradients (in Equation 5), which stems from the fact that we use aggregate statistics of node outputs from previous time steps where the model parameters were different. First, we derive the estimation error bounds for the aggregate statistics $\widehat{\mathcal{F}}_{ij}$ and $\nabla\widehat{\mathcal{F}}_{ij}$ (Lemma 4). Using these results, we bound the estimation error of fairness gradients $\nabla_\Theta H_\delta(\widehat{\mathcal{F}}_{ij})$ in the following lemma (proof in Appendix A.4).

**Lemma 3** (Fairness Gradient Estimation Error). *For a soft-routed oblique decision tree $f(\mathbf{x})$ with $L_g$-smooth activation function $g(\cdot)$, bounded i.i.d. input instances $\|\mathbf{x}_t\| \leq B$, and compact parameter set $\Theta_t \in \mathcal{B}_\mathcal{F}(0, R)$ (Frobenius norm ball of radius $R$), the estimation error can be bounded:*

$$\|\nabla_\Theta H_\delta(\mathcal{F}_{ij}) - \nabla_\Theta H_\delta(\widehat{\mathcal{F}}_{ij})\| \leq \delta B/2, \tag{7}$$

*where $\delta$ is the Huber loss parameter (Equation 4).*

Next, we use the above bound to derive the convergence of biased gradients building on the results from Ajalloeian & Stich (2020) to obtain the following result (proof in Appendix A.5):

**Theorem 1** (Gradient Norm Convergence). *Using the assumptions in Lemma 3, the expected gradient norm $\Psi_T = 1/T \sum_{t=0}^{T-1} \mathbb{E}[\|\widehat{G}(\Theta_t)\|^2]$ can be bounded as: $\Psi_T \leq \left(\epsilon + 2^{2h-2}\lambda^2\delta^2 B^2\right)$, for large enough time step $T \geq \max\left(\frac{4FL(M+1)}{\epsilon}, \frac{4FL\sigma^2}{\epsilon^2}\right)$, small step size $\gamma \leq \min\left(\frac{1}{(M+1)L}, \frac{\epsilon}{2L\sigma^2}\right)$ and $\epsilon > 0$ (see definitions in Appendix A.5).*

The above bound demonstrates that the expected norm of the gradients estimated using the aggregate statistics of decisions from previous time steps converges over time (see Appendix A.5). In the following sections, we perform experiments to empirically verify the theoretical results.

## 5 EXPERIMENTS

In this section, we present the details of our experimental setup and results to evaluate *Aranyani*. Our implementation is available at: https://github.com/brcsomnath/Aranyani/.

**Baselines**. We compare *Aranyani* ▽ with the following online learning algorithms: ■ Hoeffding Trees (HT) (Domingos & Hulten, 2000) performs decision tree learning for online data streams by leveraging the Hoeffding bound (Hoeffding, 1994), ⬠ Adaptive Hoeffding Trees (AHT) (Bifet & Gavalda, 2009) improves upon HTs by detecting changes in the input data stream and updating the learning process accordingly, ▷ FAHT (Zhang & Ntoutsi, 2019) modifies the HT splitting algorithm by introducing group fairness constraints while computing the Hoeffding bound, ⬡ MLP *Aranyani*, (MLP), uses an MLP as $f(\mathbf{x})$ and the same online learning updates described in Section 3.3 (more details about the architecture in Appendix D), ◁ Leaf *Aranyani* (Leaf) stores the aggregate gradient statistics w.r.t. leaf-level predictions or the final output instead of the node predictions, ⬤ Majority is a post-processing baseline considers the output of $f(\mathbf{x})$ with probability $p$ and outputs the majority

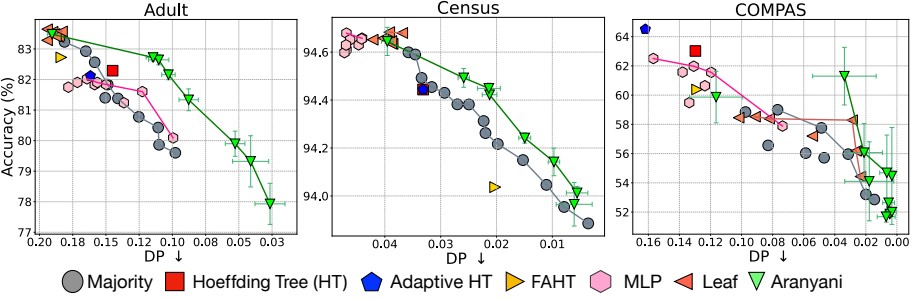

Figure 2: We report the group fairness (*demographic parity*) vs. task performance (*accuracy*) trade-off plots for different systems in (left) Adult, (center) Census, and (right) COMPAS datasets. Ideally, a fair online system should achieve low demographic parity along with high accuracy scores. Considering the inverted $x$-axis, the performance of a fair system should lie in the top right quadrant of each plot. We report *Aranyani*'s performance for different $\lambda$'s and observe that it achieves better accuracy-fairness trade-off compared to baseline systems.

label with $(1 - p)$ probability. We report the results for different values of $p$. The majority baseline can provide a fairness improvement by simply decreasing the task performance, but it requires prior access to target label information. In practice, outperforming the majority baseline is not easy. As pointed out by (Lowy et al., 2022), many offline techniques fall short of the majority's performance when batch sizes are small, which is consistent with the online learning setup.

**Online Setup & Evaluation**. We use the online learning setup described in Section 3.1. For each algorithm, we retain predictions $\hat{y}_t$ from every step and report the average task performance (accuracy) and demographic parity. In all experiments (unless otherwise stated), we use oblique forests with 3 trees and each tree has a height of 4 (based on a hyperparameter grid search). We provide further details in Appendix D. Next, we present the evaluation results of fair online learning using *Aranyani* and other baselines on 3 tabular datasets, a vision, and a language-based dataset.

## 5.1 TABULAR DATASETS

We conducted our experiments on the following tabular datasets: (a) UCI Adult (Becker & Kohavi, 1996), (b) Census (Dua et al., 2017), and (c) ProPublica COMPAS (Angwin et al., 2016). Adult dataset contains 14 features of ∼48K individuals. The task involves predicting whether the annual income of an individual is more than $50K or not and the protected attribute is gender. Census dataset has the same task description but contains 41 attributes for each individual and 299K instances. Propublica COMPAS considers the binary classification task of whether a defendant will re-offend with the protected attribute being race. COMPAS has ∼7K instances.

Figure 2 demonstrates the fairness-accuracy trade-off, where $x$ and $y$-axis show the average demographic parity (DP) and accuracy scores respectively. Ideally, we want an online system to achieve low DP and high accuracy, making the top-right quadrant the desired outcome ($x$-axis is inverted). We report *Aranyani*'s performance for different $\lambda$ values (Equation 5), which controls the trade-off. We observe that ▽ *Aranyani*'s results lie in the top-right portion, showcasing that it can achieve the best trade-off. We also observe that the variance in the results of *Aranyani* is high in COMPAS. This could be because COMPAS has fewer instances than other datasets, potentially affecting convergence. We also compare with FERMI Lowy et al. (2022), the only stochastic algorithm known to us that can be applied in online settings, and observe significant gains (Appendix E, Figure 11).

We also assess the significance of the tree structure in *Aranyani* by examining the *Aranyani* MLP ⬡ baseline that employs the same gradient accumulation method discussed in Section 3.3, but without the tree structure. In all datasets, we observe that the MLP baseline is unable to improve the fairness scores beyond a certain point. Upon further investigation, we found that this phenomenon happens due to the vanishing fairness gradients in the layers further from the output. We also observe the same phenomenon for ◁ Leaf baseline, where the fairness gradients for nodes away from the leaves become very small and they cannot be trained effectively. This showcases the importance of parameter isolation in *Aranyani* and the application of group fairness constraints on local decisions. We also note that the conventional Hoeffding tree-based baselines (HT ■, AHT ⬟, and FAHT ▷) achieve poor fairness scores, often falling behind *majority* post-processing, which shows that HT based ap-

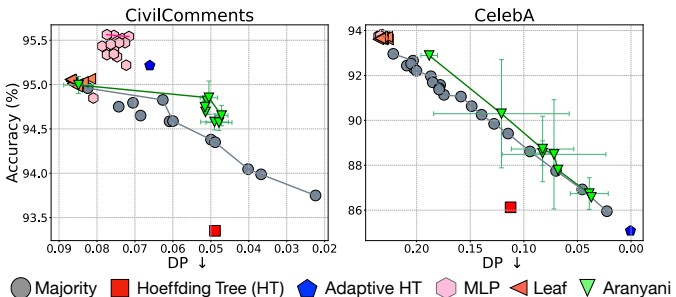

Figure 3: We report the group fairness vs. accuracy trade-off plots for different systems in (left) CivilComments and (right) CelebA datasets. We observe that *Aranyani* achieves significantly better accuracy-fairness trade-off than baseline systems.

proaches are unable to robustly improve the group fairness. Overall, we observe that *Aranyani* can achieve a better accuracy-fairness trade-off than baseline approaches across all datasets.

## 5.2 VISION & LANGUAGE DATASETS

We also conduct experiments on (a) CivilComments (Do, 2019) toxicity classification and (b) CelebA (Liu et al., 2015) image classification datasets. CivilComments is a natural language dataset where the task involves classifying whether an online comment is toxic or not. We use religion as the protected attribute and consider instances of religion labels: "*Muslim*" and "*Christian*", as they showcase the maximum discrepancy in toxicity. For CivilComments, we obtain text representations from the instruction-tuned model, Instructor (Su et al., 2023) by using the prompt "Represent a toxicity comment for classifying its toxicity as toxic or non-toxic: [`comment`]". CelebA dataset contains 200K images of celebrity faces with 40 categorical attributes. Following previous works (Jung et al., 2022b; Qiao & Peng, 2022; Jung et al., 2022a; Liu et al., 2021), we select "blond hair" as the task label and "gender" as the protected attribute. For CelebA, we retrieve the image representations from the CLIP model (Radford et al., 2021).

In Figure 3, we report the fairness-accuracy trade-off plots, where we observe that *Aranyani* ▽ achieves the best accuracy-fairness trade-off on both datasets. Similar to tabular datasets, we observe that the MLP ⬠ and Leaf ◁ baselines are unable to improve the fairness scores at all. Hoeffding tree (HT) baseline ■ achieves decent fairness scores but is unable to converge on the task. This highlights the limitations of traditional decision trees when dealing with non-axis-aligned data.

## 5.3 ANALYSIS

In this section, we perform empirical evaluations to analyze the functioning of *Aranyani*.

**Reservoir Variant**. We compare the performance of *Aranyani* with a variant (using oblique forests) that stores all samples in the online stream to compute the fairness loss. We refer to this variant as "*Reservoir*". In Figure 4 (left), we observe that *Aranyani* achieves a similar accuracy-fairness tradeoff compared to the reservoir variant on the Adult dataset. As *Aranyani* does not need to store previous input samples, it is quite efficient – achieving a ~3x improvement in computation time and ~23% reduction in memory utilization.

**Gradient Convergence**. We investigate the convergence of the gradients used to update the oblique tree. We conduct our experiment on CivilComments dataset using *Aranyani* with a single tree and report the norm of the total gradients and fairness gradients used to update $\mathbf{W}$ (weight parameter of each node). In Figure 4 (center), we observe that the norm of both task and fairness gradients converge over time during the online learning process. This corroborates our theoretical guarantees in Lemma 1. We found this behavior to be consistent across different parameters (Appendix E).

**Tree Height Ablations**. We study the effect of varying the tree height in oblique forests on the fairness-accuracy tradeoff. In Figure 4 (right), we report *Aranyani*'s performance on the Adult dataset with a fixed parameter $\lambda = 0.1$. We observe that the accuracy increases with height and the demographic parity worsens ($y$-axis is inverted) with increasing height. This is consistent with our theoretical results (Lemma 1 & 2). However, we have observed a slight decrease in accuracy when using large tree heights (height=8). This observation suggests that oblique trees may overfit

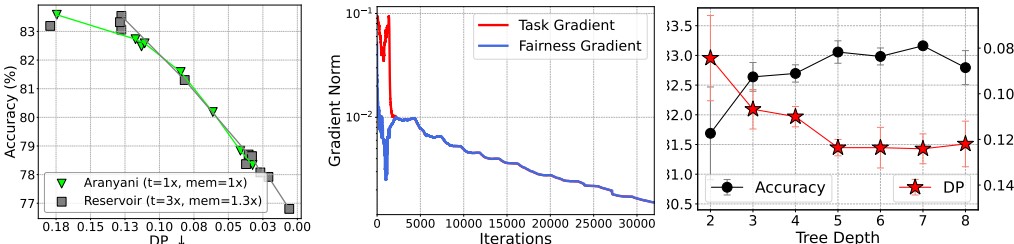

Figure 4: (left) We compare *Aranyani* with the reservoir variant that stores all input instances, (center) we investigate the gradient convergence, and (right) the impact of tree height on performance.

the training data when reaching a certain height, and choosing a shallower tree could be beneficial. We perform additional experiments to investigate the performance of *Aranyani* (see Appendix E).

## 6 RELATED WORKS

**Fairness.** Existing work on promoting group fairness can be classified into three categories: pre-processing (Zemel et al., 2013; Calmon et al., 2017), post-processing (Hardt et al., 2016; Pleiss et al., 2017; Alghamdi et al., 2022), and in-processing (Quadrianto & Sharmanska, 2017; Agarwal et al., 2018; Mary et al., 2019; Prost et al., 2019; Baharlouei et al., 2019; Lahoti et al., 2020; Lowy et al., 2022; Basu Roy Chowdhury et al., 2021; Chowdhury & Chaturvedi, 2022; Chowdhury et al., 2023; Baharlouei et al., 2024) techniques. These are trained and tested on a static dataset, and there is a growing concern that they fail to remain fair under distribution shifts (Barrett et al., 2019; Mishler & Dalmasso, 2022; Rezaei et al., 2021; Wang et al., 2023). This calls for fairness-aware systems that can adapt to distribution changes and update themselves in an online manner. However, an online learning setting, where input instances arrive one at a time, presents several challenges that make it difficult to apply existing techniques: (a) pre-processing techniques are not feasible as they require prior access to input instances; (b) post-processing techniques often require access to a held-out set, which may be impractical; and (c) in-processing techniques need a batch of samples to estimate the group fairness loss, which may not always be feasible due to privacy reasons (Voigt & Von dem Bussche, 2017) or distributed infrastructure (Konečný et al., 2016). FERMI (Lowy et al., 2022) is the only in-processing approach that can work with a batch size of 1. Although several works (Gillen et al., 2018; Bechavod et al., 2020) have studied individual fairness in online settings, only a few recent works (Chowdhury & Chaturvedi, 2023; Zhao et al., 2023; Chen et al., 2023; Yin et al., 2024; Truong et al., 2024; Jiang et al., 2024) considered group fairness in settings where the underlying task or data distribution changes over time. However, these systems are incrementally trained on sub-tasks, requiring access to task labels, which is not available in online data streams.

**Gradient-based learning of trees.** Similar to *Aranyani* that leveraged a gradient-based objective, (Karthikeyan et al., 2022) uses gradient-based methods to discover the structure of oblique decision tree classifiers. Discovering tree structures with gradient-based methods has been also considered in works on autoencoders (Nagano et al., 2019; Shin et al., 2016), on discovering structure (e.g., parses) in natural language (Yin et al., 2018; Drozdov et al., 2019), hierarchical clustering (Monath et al., 2017; 2019; Zhao et al., 2020; Chami et al., 2020), phylogenetics (Macaulay & Fourment, 2023; Penn et al., 2023), and extreme classification (Yu et al., 2022; Jernite et al., 2017; Sun et al., 2019; Jasinska-Kobus et al., 2021). We discuss more prior works related to decision trees in Appendix B.

## 7 CONCLUSION

In this paper, we propose *Aranyani*, a framework to achieve group fairness in online learning. *Aranyani* uses an ensemble of oblique decision trees and leverages its hierarchical prediction structure to store aggregate statistics of local decisions. These aggregate statistics help in the efficient computation of group fairness gradients in the online setting, eliminating the need to store previous input instances. Empirically, we observe that *Aranyani* achieves significantly better accuracy-fairness trade-off compared to baselines on a wide range of tabular, image, and text classification datasets. Through extensive analysis, we showcase the utility of our proposed tree-based prediction structure and fairness gradient approximation. While we investigated binary oblique tree structures, their ability to fit complex functions can be limited when compared to fully connected networks. Future research can explore other parameterizations (e.g., graph-based structures) that enable effective gradient computation to impose group fairness in online settings with superior prediction power.

## REPRODUCIBILITY STATEMENT

We have submitted the implementation of *Aranyani* in the supplementary materials. We have extensively discussed the details of our experimental setup, datasets, baselines, and hyperparameters in Section 5 and Appendix E. We have also provided the details of our training procedure in Section 3.4 and Appendix D.3. We will make our implementation public once the paper is published.

## ETHICS STATEMENT

In this paper, we introduce an online learning framework, *Aranyani*, to enhance group fairness while making decisions for an online data stream. *Aranyani* has been developed with the intention of mitigating biases of machine learning systems when they are deployed in the wild. However, it is crucial to thoroughly assess the data's quality and the accuracy of the demographic labels used for training *Aranyani*, as otherwise, it may still encode negative biases. In our experiments, we use publicly available datasets and obtain data representations from open-sourced models. We do not obtain any demographic labels through data annotation or any private sources.

## ACKNOWLEDGEMENTS

The authors are thankful to James Atwood, Anneliese Brei, Shounak Chattopadhyay, Haoyuan Li, and Anvesh Rao Vijjini for helpful feedback and discussions. The work began when Somnath Basu Roy Chowdhury was a student researcher at Google. Somnath Basu Roy Chowdhury and Snigdha Chaturvedi were partly supported by Amazon Research Awards and the National Science Foundation under award DRL-2112635.

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

# A  THEORETICAL PROOFS

CONTENTS

## A.1   ASSUMPTIONS

In this section, we will present standard regularity assumptions utilized in deriving the theoretical results. We derive the results for a soft-routed oblique decision forest, $f(\mathbf{x})$, with a single tree, $|\mathcal{T}| = 1$. However, these can be easily extended to a setup with multiple trees. Specifically, we derive the estimation errors in gradients leveraging the framework of Ajalloeian & Stich (2020), i.e., we assume our stochastic gradient estimation for the objective has the following form: $\widehat{G}(\Theta, \xi) = G(\Theta) + b(\Theta) + n(\xi)$, where $b(\Theta)$ and $n(\xi)$ denotes the bias and noise involved in the estimation. We present an exact description of all the assumptions used in our setup and optimization process:

(A1)  The input instances $\mathbf{x}_t$ arrive in an i.i.d. fashion and are bounded: $\|\mathbf{x}_t\| < B$.

(A2)  The parameter set is compact and lies within a Frobenius ball $\Theta \in \mathcal{B}_F(0, R)$ of radius $R$.

(A3)  $f(\mathbf{x})$ denotes a soft-routed binary oblique forest with a single tree ($|\mathcal{T}| = 1$), activation function, $g(\cdot)$, and leaf parameters $\|\theta_l\| = 1$.

(A4)  The activation function $g(\cdot)$ used in oblique trees is $L_g$ smooth.

(A5)  The noise in gradient estimation $n(\xi)$ has zero mean $\mathbb{E}_\xi[n(\xi)] = \mathbf{0}$ and is $(M, \sigma^2)$ bounded:

$$\mathbb{E}_\xi[\|n(\xi)\|^2] \le M\|G(\Theta) + b(\Theta)\|^2 + \sigma^2, \forall \Theta. \tag{8}$$

(A6)  The task loss function $\mathcal{L}(\cdot, \cdot)$ is $K_t$-Lipschitz.

## A.2   PROOF OF LEMMA 1

First, we present a proposition that would be helpful in proving Lemma 1.

**Proposition 1.** *Let $p_i$ and $q_i$ be samples from a random variable $P$ that is bounded between $[0, 1]$. Then, the following inequality holds:*

$$\left| \prod_i p_i - \prod_i q_i \right| \le \sum_i |q_i - p_i|. \tag{9}$$

*Proof.* The proof is presented below:

$$\begin{aligned}
\left| \prod_i p_i - \prod_i q_i \right| &\le \prod_i \max\{p_i, q_i\} - \prod_i \min\{p_i, q_i\} \\
&\le \sum_i |q_i - p_i| \prod_{j \ne i} \max\{p_j, q_j\} \\
&\le \sum_i |q_i - p_i|.
\end{aligned}$$

$\square$

Next, we proceed to the main proof of Lemma 1.

*Proof of Lemma 1.* The proof is completed using the following steps:

$$\text{DP} = |\mathbb{E}[f(\mathbf{x}|a=0)] - \mathbb{E}[f(\mathbf{x}|a=1)]|$$

$$= \left| \mathbb{E}\left[ \sum_l p_l(\mathbf{x}|a=0)\theta_l \right] - \mathbb{E}\left[ \sum_l p_l(\mathbf{x}|a=1)\theta_l \right] \right|$$

$$= \left| \sum_l \left( \mathbb{E}\left[ p_l(\mathbf{x}|a=0) \right] - \mathbb{E}\left[ p_l(\mathbf{x}|a=1) \right] \right) \theta_l \right|$$

$$\leq \sum_l \left| \mathbb{E}\left[ p_l(\mathbf{x}|a=0) \right] - \mathbb{E}\left[ p_l(\mathbf{x}|a=1) \right] \right| |\theta_l|$$

$$= \sum_l \left| \mathbb{E}\left[ \prod_i n_{i,A(i,l)}(\mathbf{x}|a=0) \right] - \mathbb{E}\left[ \prod_i n_{i,A(i,l)}(\mathbf{x}|a=1) \right] \right|$$

$$= \sum_l \left| \frac{1}{m} \sum_{\mathbf{x}_0 \sim p(\mathbf{x}|a=0)} \left[ \prod_i n_{i,A(i,l)}(\mathbf{x}_0) \right] - \frac{1}{m} \sum_{\mathbf{x}_1 \sim p(\mathbf{x}|a=1)} \left[ \prod_i n_{i,A(i,l)}(\mathbf{x}_1) \right] \right|$$

$$= \sum_l \left| \frac{1}{m} \sum_{\mathbf{x}_0 \sim p(\mathbf{x}|a=0), \mathbf{x}_1 \sim p(\mathbf{x}|a=1)} \left( \prod_i n_{i,A(i,l)}(\mathbf{x}_0) - \prod_i n_{i,A(i,l)}(\mathbf{x}_1) \right) \right|$$

$$\leq \sum_l \frac{1}{m} \sum_{\mathbf{x}_0,\mathbf{x}_1} \sum_i \left| n_{i,A(i,l)}(\mathbf{x}_0) - n_{i,A(i,l)}(\mathbf{x}_1) \right| \tag{10}$$

$$= \sum_l \sum_i \mathbb{E}\left[ \left| n_{i,A(i,l)}(\mathbf{x}_0) - n_{i,A(i,l)}(\mathbf{x}_1) \right| \right] \tag{11}$$

$$\leq \sum_l \sum_i \epsilon$$

$$= h2^h \epsilon.$$

In the above proof, the first few steps use the linearity of expectation. We derive Equation 10 by using the result in Proposition 1. Finally, plugging the bound from Equation 6 in Equation 11 we obtain the final result. $\square$

## A.3 PROOF OF LEMMA 2

First, we present the definition of empirical Rademacher complexity, $\hat{R}_n(\mathcal{H})$, for function class, $\mathcal{H}$.

$$\hat{R}_n(\mathcal{H}) := \frac{1}{n}\mathbb{E}_\epsilon \left[ \sup_{h \in \mathcal{H}} \sum_{t=1}^n \epsilon_t f(\mathbf{x}_t) \right]. \tag{12}$$

Intuitively, the above definition measures the expressivity of a function class $\mathcal{H}$ by measuring the ability to fit random noise. In the above equation, for a given set $S = \{\mathbf{x}_1, \ldots, \mathbf{x}_n\}$ and Rademacher vector $\epsilon$, the supremum quantifies the maximum correlation between $f(\mathbf{x}_t)$ and randomly generated samples $\epsilon_t \in \{-1, 1\} \, \forall f \in \mathcal{H}$.

*Proof.* In Equation 12, we plug in the soft-routed binary oblique decision tree formulation, $f(\mathbf{x}) = \sum_{l=1}^{2^h} \prod_{i=1}^h g(\mathbf{w}_{i,A(i,l)}^T \mathbf{x} + \mathbf{b}_{i,A(i,l)})\theta_l$, in the above equation to obtain:

$$\hat{R}_n(\mathcal{H}) = \frac{1}{n}\mathbb{E}_\epsilon\left[\sup_{h\in\mathcal{H}}\sum_{t=1}^{n}\epsilon_t\sum_{l=1}^{2^h}\prod_{i=1}^{h}g(\mathbf{w}_{i,A(i,l)}^T\mathbf{x}_t+\mathbf{b}_{i,A(i,l)})\theta_l\right]$$

$$\leq \frac{2^h}{n}\mathbb{E}_\epsilon\left[\sup_{h\in\mathcal{H}}\max_l\sum_{t=1}^{n}\epsilon_t\prod_{i=1}^{h}g(\mathbf{w}_{i,A(i,l)}^T\mathbf{x}_t+\mathbf{b}_{i,A(i,l)})\theta_l\right]$$

$$\leq \frac{2^h}{n}\mathbb{E}_\epsilon\left[\sup_{h\in\mathcal{H}}\sum_{t=1}^{n}\epsilon_t\max_l\prod_{i=1}^{h}g(\mathbf{w}_{i,A(i,l)}^T\mathbf{x}_t+\mathbf{b}_{i,A(i,l)})\theta_l\right].$$

Next, we continue the proof by using $F(\mathbf{x}_t) = \max_l\prod_{i=1}^{h}g(\mathbf{w}_{i,A(i,l)}^T\mathbf{x}_t+\mathbf{b}_{i,A(i,l)})\theta_l$ and $\|\theta_l\| = 1$.

$$\hat{R}_n(\mathcal{H}) \leq \frac{2^h}{n}\mathbb{E}_\epsilon\left[\sup_{h\in\mathcal{H}}\sum_{t=1}^{n}\epsilon_t F(\mathbf{x}_t))\right]$$

$$\leq \frac{2^h}{n}\sqrt{\sup_{h\in\mathcal{H}}\mathbb{E}_\epsilon\left[\left\|\sum_{t=1}^{n}\epsilon_t F(\mathbf{x}_t)\right\|^2\right]}$$

$$= \frac{2^h}{n}\sqrt{\sup_{h\in\mathcal{H}}\mathbb{E}_\epsilon\left[\sum_{t=1}^{n}\epsilon_t^2 F(\mathbf{x}_t)^2\right]+\mathbb{E}_\epsilon\left[\sum_{t\neq t'}\epsilon_t'\epsilon_t F(\mathbf{x}_t)F(\mathbf{x}_t')\right]}$$

$$= \frac{2^h}{n}\sqrt{\mathbb{E}_\epsilon\left[\sum_{t=1}^{n}\epsilon_t^2\right]}$$

$$\leq 2^h/\sqrt{n}.$$

$\square$

## A.4  PROOF OF LEMMA 3

The proof of Lemma 3 builds on the intermediate results that we present in the sequel. First, we derive the bound on the estimation error in $\nabla\widehat{\mathcal{F}}_{ij}$. Using this result, we can bound the estimation in the fairness gradients derived in Equation 5.

For simplicity of notation, in the subsequent proof, we denote input instances from different demographic groups as $\mathbf{x}_i \sim p(\mathbf{x}|a=0)$ and $\mathbf{x}_j \sim p(\mathbf{x}|a=1)$. With a slight abuse of notation, we also use the indices $i, j$ to indicate the time step at which the instance $\mathbf{x}_i$ arrives.

**Lemma 4** (Estimation error in $\nabla\mathcal{F}_{ij}$). *Under assumptions (A1), (A2), (A3), (A4), the estimation error of $\nabla\mathcal{F}_{ij}$ is bounded as:* $|\nabla\mathcal{F}_{ij} - \nabla\widehat{\mathcal{F}}_{ij}| \leq \min\{B/4, 2L_g R\}$.

*Proof.* Next, we focus on approximating the estimation error in $\nabla\widehat{\mathcal{F}}_{ij}$.

$$\nabla\mathcal{F}_{ij} = \mathbb{E}_i[\nabla n(\Theta_t, \mathbf{x}_i)] - \mathbb{E}_j[\nabla n(\Theta_t, \mathbf{x}_j)],\ \nabla\widehat{\mathcal{F}}_{ij} = \mathbb{E}_i[\nabla n(\Theta_i, \mathbf{x}_i)] - \mathbb{E}_j[\nabla n(\Theta_j, \mathbf{x}_j)].$$

Similarly, we can estimate the approximation error as:

$$|\nabla\mathcal{F}_{ij} - \nabla\widehat{\mathcal{F}}_{ij}| = |\mathbb{E}_i[\nabla n(\Theta_t, \mathbf{x}_i)] - \mathbb{E}_i[\nabla n(\Theta_i, \mathbf{x}_i)] - \mathbb{E}_j[\nabla n(\Theta_t, \mathbf{x}_j)] + \mathbb{E}_j[\nabla n(\Theta_j, \mathbf{x}_j)]|$$
$$= |\mathbb{E}_i[\nabla n(\Theta_t, \mathbf{x}_i) - \nabla n(\Theta_i, \mathbf{x}_i)] - \mathbb{E}_j[\nabla n(\Theta_t, \mathbf{x}_j) - \nabla n(\Theta_j, \mathbf{x}_j)]|$$
$$\leq |\mathbb{E}_i[\nabla n(\Theta_t, \mathbf{x}_i) - \nabla n(\Theta_i, \mathbf{x}_i)]| + |\mathbb{E}_j[\nabla n(\Theta_t, \mathbf{x}_j) - \nabla n(\Theta_j, \mathbf{x}_j)]|$$
$$= L_g\mathbb{E}_i[\|\Theta_t - \Theta_i\|] + L_g\mathbb{E}_j[\|\Theta_t - \Theta_j\|]$$
$$= L_g\mathbb{E}_i[\|\Theta_t - \Theta_i\|]$$
$$\leq 2L_g R,$$

where $L_g$ is the smoothness constant of the activation function. Note that this error bound doesn't increase in an unrestricted manner as the node gradients are bounded as:

$$|\nabla n(\Theta_t, \mathbf{x})| \leq \|\mathbf{x}\|/4 \leq B/4.$$

Therefore, we can derive a tighter bound as:

$$|\nabla \mathcal{F}_{ij} - \nabla \widehat{\mathcal{F}}_{ij}| \leq \min\left\{\frac{B}{4}, 2L_g R\right\}.$$

For sigmoid activation, we can plug in $L_g = \frac{2\sqrt{3}-3}{18}$ in the above equation to get the exact bound. $\quad\square$

Using the bounds derived in the above lemmas, we proceed towards the proof of Lemma 3.

*Proof of Lemma 3.* We begin by noting that the task loss gradients are unbiased as the input instance $\mathbf{x}_t$ is sampled in an i.i.d. fashion making the gradient estimate unbiased on expectation.

$$\|\nabla_\Theta H_\delta(\mathcal{F}_{ij}) - \nabla_\Theta H_\delta(\widehat{\mathcal{F}}_{ij})\| =$$
$$\begin{cases} \|\mathcal{F}_{ij}.\nabla_\Theta \mathcal{F}_{ij} - \widehat{\mathcal{F}}_{ij}\nabla_\Theta \widehat{\mathcal{F}}_{ij}\|, & \text{if } |\widehat{\mathcal{F}}_{ij}| < \delta \\ \delta\|\text{sgn}\left(\mathcal{F}_{ij} - \delta/2\right)\nabla_\Theta \mathcal{F}_{ij} - \text{sgn}\left(\widehat{\mathcal{F}}_{ij} - \delta/2\right)\nabla_\Theta \widehat{\mathcal{F}}_{ij}\|, & \text{otherwise} \end{cases}.$$

Therefore, the approximation error arises from the fairness gradient terms. First, we consider the approximation error in the gradients when $|\widehat{\mathcal{F}}_{ij}| \leq \delta$. It can be written as:

$$|\mathcal{F}_{ij}\nabla \mathcal{F}_{ij} - \widehat{\mathcal{F}}_{ij}\nabla \widehat{\mathcal{F}}_{ij}| \leq \delta|\nabla \mathcal{F}_{ij} - \nabla \widehat{\mathcal{F}}_{ij}| \leq \delta B/4.$$

Note that in the above equation, we consider the weaker upper bound for the estimation error of $\nabla \mathcal{F}_{ij}$ of $B/4$. As the bound on the input $\|\mathbf{x}\| \leq B$ is easier to work with. Next, we consider the case where $|\mathcal{F}_{ij}| > \delta$:

$$\delta|\text{sgn}(\mathcal{F}_{ij} - \delta/2)\nabla \mathcal{F}_{ij} - \text{sgn}(\widehat{\mathcal{F}}_{ij} - \delta/2)\nabla \widehat{\mathcal{F}}_{ij}| \leq \delta|2\nabla \mathcal{F}_{ij}| \leq \delta B/2.$$

Therefore, we observe that the overall error can be bounded as:

$$\|\nabla_\Theta H_\delta(\mathcal{F}_{ij}) - \nabla_\Theta H_\delta(\widehat{\mathcal{F}}_{ij})\| \leq \delta B/2. \tag{13}$$

$\square$

## A.5 PROOF OF THEOREM 1

**Theorem 2** (Precise Statement of Theorem 1). *Using the assumptions (A1), (A2), (A3), (A4), (A5), for $\epsilon > 0$ the expected gradient norm $\Psi_T = 1/T \sum_{t=0}^{T-1} \mathbb{E}[\|\widehat{G}(\Theta_t)\|^2]$ can be bounded as $\Psi_T \leq \left(\epsilon + 2^{2h-2}\lambda^2\delta^2 B^2\right)$ for*

$$T \geq \max\left(\frac{4FL(M+1)}{\epsilon}, \frac{4FL\sigma^2}{\epsilon^2}\right) \text{ and } \gamma \leq \min\left(\frac{1}{(M+1)L}, \frac{\epsilon}{2L\sigma^2}\right). \tag{14}$$

*where $F = \mathbb{E}[\mathcal{L}_o(\theta_t)] - \mathcal{L}_o^*$ with $\mathcal{L}_o$ denoting the overall loss function (Equation 4).*

*Proof.* Due to estimation errors in the online setting, the gradients we use are biased and can be written as:

$$\widehat{G}(\Theta, \xi) = G(\Theta) + b(\Theta) + n(\xi) \tag{15}$$

where $G(\Theta)$ is the exact gradient (computed using all the input instances seen so far), $b(\Theta)$ is the bias or estimation error, and $n(\xi)$ is the noise coming from the i.i.d. estimate of the loss function. From Lemma 3, we can bound the overall estimation bias as:

$$\|b(\Theta)\| \leq \lambda \sum_{\forall i,j} \|\nabla_\Theta H_\delta(\mathcal{F}_{ij}) - \nabla_\Theta H_\delta(\widehat{\mathcal{F}}_{ij})\| \leq 2^{h-1}\lambda\delta B. \tag{16}$$

and $n(\xi)$ has zero mean. We assume that the noise $n(\xi)$ is $(M, \sigma^2)$ bounded as defined by Ajalloeian & Stich (2020). Note that $M$ depends on the task loss function $\mathcal{L}(\cdot, \cdot)$ and is not a function of $t$. We use the result from Lemma 2 in (Ajalloeian & Stich, 2020) to obtain:

$$\Psi_T \leq \frac{2F}{T\gamma} + 2^{2h-2}\lambda^2\delta^2 B^2 + \gamma L\sigma^2 \tag{17}$$

where we denote the average gradient norm as: $\Psi_T = \frac{1}{T}\sum_{t=0}^{T-1} \mathbb{E}[\|\widehat{G}(\Theta_t)\|^2]$, $F = \mathbb{E}[\mathcal{L}_o(\Theta_0)] - \mathcal{L}_o^*$ with $\mathcal{L}_o$ denoting the overall loss function, and $\gamma$ is the step size. We assume that $\mathcal{L}_o$ has a smoothness constant of $L$.

Then, for $\epsilon > 0$ we can show that the expected gradient norm converges as follows:

$$\Psi_T \leq \frac{\epsilon}{2} + \frac{\epsilon}{2} + 2^{2h-2}\lambda^2\delta^2 B^2 = \epsilon + 2^{2h-2}\lambda^2\delta^2 B^2. \tag{18}$$

for large enough time step (or input samples) and small enough step size:

$$T \geq \max\left(\frac{4FL(M+1)}{\epsilon}, \frac{4FL\sigma^2}{\epsilon^2}\right) \text{ and } \gamma \leq \min\left(\frac{1}{(M+1)L}, \frac{\epsilon}{2L\sigma^2}\right), \tag{19}$$

which completes the proof. $\qquad\square$

**Discussion**. Theorem 2 shows that the convergence of the gradient norms depends exponentially on the tree depth, $O(2^{2h})$. However, we used shallow trees ($h \leq 10$) and observed that shallow trees can provide a good accuracy-fairness tradeoff. In Appendix E, we empirically study the gradient norm at the end of online training and how it varies with the tree height. Surprisingly, we observe a linear correlation between the gradient norm and the tree height for small $h \leq 10$. The experiment yielded consistently small gradient norms that had no discernible impact on the final accuracy or DP results, which shows that the gradient estimation process works in practice. Theoretically explaining this phenomenon for small $h$ is non-trivial and we leave it for future works to explore this result.

## A.6 ADDITIONAL THEORETICAL ANALYSIS

In this section, we provide additional theoretical results analyzing the properties of *Aranyani*. Specifically, we derive the gradient bound for $G(\Theta)$ (Equation 5) and show the oblique decision trees are $K_f$-Lipschitz continuous and $L_f$-smooth.

**Lemma 5** (Bounded gradients). *Under assumptions (A1), (A3), (A6), and activation function $g(\cdot)$ is a sigmoid function the gradients are bounded as:*

$$\|G(\Theta)\| \leq K_t + 2^{h-2}\lambda\delta B, \tag{20}$$

*where $\delta$ is the Huber parameter and $\lambda$ is a hyperparameter (Equation 5).*

*Proof.* Using Equation 5, the gradient bound can be written as:

$$\|G(\Theta)\| \leq \max \left( \|\nabla \mathcal{L}(f(\mathbf{x}), y)\| + \lambda \sum_{\forall i,j} \|\nabla H_\delta(\mathcal{F}_{ij})\|, \right)$$

$$\leq K_t + \lambda \sum_{\forall i,j} \max(\|\mathcal{F}_{ij}\| \|\nabla \mathcal{F}_{ij}\|, \delta \|\nabla \mathcal{F}_{ij}\|) \tag{21}$$

$$\leq K_t + \lambda \delta \sum_{\forall i,j} \max \|\nabla \mathcal{F}_{ij}\|$$

$$\leq K_t + \lambda \delta 2^h \max \|\mathbb{E}[\nabla n_{ij}(\mathbf{x}|a=0)] - \mathbb{E}[\nabla n_{ij}(\mathbf{x}|a=1)]\|$$

$$\leq K_t + \lambda \delta 2^h \max \|\mathbb{E}[\nabla n_{ij}(\mathbf{x}|a=0)]\|$$

$$= K_t + \lambda \delta 2^h \max \|\mathbb{E}[n_{ij}(\mathbf{x}|a=0)(1 - n_{ij}(\mathbf{x}|a=0))\mathbf{x}]\| \tag{22}$$

$$\leq K_t + \lambda \delta 2^h \max \|n_{ij}(\mathbf{x}|a=0)(1 - n_{ij}(\mathbf{x}|a=0))\| \|\mathbf{x}\|$$

$$= K_t + 2^{h-2} \lambda \delta B.$$

In Equation 21, the first term is upper bounded by $\delta \|\nabla \mathcal{F}_{ij}\|$ because the gradient is selected only when $|\mathcal{F}_{ij}| \leq \delta$. The maxima in Equation 22 is achieved when $n_{ij}(\mathbf{x}) = 0.5$. For cross-entropy loss with softmax, the upper bound can be derived by plugging in $K_t = \sqrt{2}$:

$$\|G(\Theta)\| \leq \sqrt{2} + 2^{h-2} \lambda \delta B. \tag{23}$$

Note that even though the gradient bound has an exponential term ($2^{h-2}$), in practice due to parameter isolation the gradients for individual node parameters have a much lower bound as each tree node is only associated with a single $\mathcal{F}_{ij}$ constraint. $\square$

**Lemma 6** (Lipschitz Continuity & Smoothness). *Under assumptions (A1), (A3), $f(\mathbf{x})$ is $K_f$-Lipschitz and $L_f$-smooth, where $K_f = 2^{h-2} hB$ and $L_f = 2^{h-4} h(h+1)B^2$.*

*Proof.* Using the mean value theorem, the Lipschitz constant of a function $f(x)$ is bounded by $\max \|\nabla f(x)\|$. Therefore, finding the upper bound on the derivative is sufficient.

$$f(\mathbf{x}, \Theta) = \sum_{i=1}^{2^h} \prod_{j=1}^{h} n_{ij}(\mathbf{x})\theta_i$$

$$\nabla_\Theta f(\mathbf{x}, \Theta) = \sum_{i=1}^{2^h} \sum_{k=1}^{h} n_{ik}(\mathbf{x})(1 - n_{ik}(\mathbf{x})) \prod_{j=1, j \neq k}^{h} n_{ij}(\mathbf{x})\theta_i \mathbf{x}$$

$$\leq \sum_{i=1}^{2^h} \sum_{k=1}^{h} \frac{\theta_i \mathbf{x}}{4}$$

$$\leq \sum_{i=1}^{2^h} \sum_{k=1}^{h} \|\theta\| \|\mathbf{x}\|/4$$

$$= 2^{h-2} h \|\theta\| \|\mathbf{x}\|$$

where $\|\theta\|$ and $\|\mathbf{x}\|$ denote the maximum norm of parameters $\theta_i$ and input $\mathbf{x}$ respectively. Next, plugging in the assumptions about the norm we get the following:

$$\nabla_\Theta f(\mathbf{x}, \Theta) \leq 2^{h-2} Bh = K_f.$$

Similarly, we can derive the expression for the smoothness constant $L_f$. Using the mean value theorem, we know that $\|\nabla_\Theta^2 f(\mathbf{x}, \Theta)\| \leq L_f$. Therefore, we derive an upper bound for the former:

$$\nabla_\Theta^2 f(\mathbf{x}, \Theta) = \sum_{i=1}^{2^h} \sum_{k=1}^{h} n_{ik}(\mathbf{x})(1 - n_{ik}(\mathbf{x}))(1 - 2n_{ik}(\mathbf{x})) \prod_{j=1, j \neq k}^{h} n_{ij}(\mathbf{x})\theta_i \mathbf{x}^2$$

$$+ \sum_{i=1}^{2^h} \sum_{k=1}^{h} n_{ik}(\mathbf{x})(1 - n_{ik}(\mathbf{x})) \sum_{m=1, m \neq k}^{h} n_{im}(\mathbf{x})(1 - n_{im}(\mathbf{x})) \prod_{j=1, j \neq \{k,m\}}^{h} n_{ij}(\mathbf{x})\theta_i \mathbf{x}^2$$

$$\leq \frac{2^h h}{6\sqrt{3}} \|\theta\| \|x\|^2 + h(h-1)2^{h-4} \|\theta\| \|\mathbf{x}\|^2$$

$$\leq 2^{h-4} h(h+1) \|\theta\| \|\mathbf{x}\|^2$$

$$= 2^{h-4} h(h+1) B^2 = L_f.$$

This completes the proof. $\square$

# B    ADDITIONAL RELATED WORK

*Aranyani*[2] uses a gradient-based approach to learn the parameters of an oblique decision tree. In this section, we discuss prior works that utilized decision trees in the online learning setting.

**Decision Trees.** Decision trees have been studied for the online setting where data arrives in a stream over time, particularly to mitigate catastrophic forgetting (Kirkpatrick et al., 2017). Initial works (Kamiran et al., 2010; Raff et al., 2018; Aghaei et al., 2019) explored various formulations of the splitting criterion to improve fairness in decision trees within the offline setting. More recent work (Zhang & Ntoutsi, 2019) in fair online learning leveraged Hoeffding trees (Domingos & Hulten, 2000) to process online data streams, and introduced group fairness constraints in its splitting criteria. However, this approach has several drawbacks: (a) conventional decision trees can only function with axis-aligned data making it unsuitable for more complex data domains like text or images, (b) it cannot be trained using gradient descent making it difficult to plug in additional modules like a text or image encoder. In contrast to these approaches, *Aranyani* leverages oblique decision trees parameterized by neural networks improving its expressiveness while making it amenable to gradient-based updates using modern accelerators. *Aranyani* exploits its tree structure to store aggregate statistics of the local node outputs to compute the fairness gradients without requiring it to store additional samples.

---

[2]The name of our approach was inspired by the Hindu goddess of forests and wild animals, *Aranyani*. She fearlessly navigated the wilderness, treating humans and animals equally. These characteristics bear resemblance to the desired traits of our system, which must be fair and functional in the wild (online settings).

# C    EXTENSIONS OF *Aranyani*

In this section, we discuss several ways to extend *Aranyani* to handle non-binary protected attribute labels or different notions of group fairness objectives.

## C.1    HANDLING DIFFERENT NOTIONS OF GROUP FAIRNESS

In this section, we show that *Aranyani* can be used to impose different notions of group fairness. Specifically, we derive the fairness constraints in *Aranyani* for the group fairness notion of *equalized odds* (Hardt et al., 2016). However, note that *Aranyani* can be extended to handle any conditional moment-based group fairness loss. For equalized odds, the node-level constraints can be shown as:

$$\mathcal{F}_{ij}^c = \mathbb{E}[n_{ij}(\mathbf{x}|y = c, a = 0)] - \mathbb{E}[n_{ij}(\mathbf{x}|y = c, a = 1)],$$

where $c$ denotes different task labels. The objective in the offline setup is formulated as:

$$\min_f \mathcal{L}(f(\mathbf{x}), y) + \lambda \sum_{i,j} \sum_c H_\delta(\mathcal{F}_{ij}^c).$$

The gradient estimation in the online setting requires the storage of the following aggregate estimates: (a) $\mathbb{E}[n_{ij}(\mathbf{x}|y = l, a = 0)]$, (b) $\mathbb{E}[\nabla_\Theta n_{ij}(\mathbf{x}|y = c, a = 0)]$, (c) $\mathbb{E}[n_{ij}(\mathbf{x}|y = c, a = 1)]$, and (d) $\mathbb{E}[\nabla_\Theta n_{ij}(\mathbf{x}|y = c, a = 1)], \forall l \in \{0, \dots, C - 1\}$. This would result in an overall storage cost of $\mathcal{O}(2^h C d)$, where $d$ is the dimension of the input $\mathbf{x}$ and $C$ is the number of task labels. Similarly, *Aranyani* can be extended to handle other group fairness notions like equality of opportunity (Hardt et al., 2016), and representation parity (Hashimoto et al., 2018) as well. We report initial results with the equality of opportunity fairness notion in Appendix E.

## C.2    HANDLING NON-BINARY PROTECTED ATTRIBUTES

In this section, we show how *Aranyani* can be extended to process protected attributes with more than two labels. Let us assume the protected label $k \in \{0, \dots, K - 1\}$. In this case, the fairness loss is defined as the difference between the expected overall output and the expected output for a specific protected group as shown below:

$$\mathcal{F}_{ij}^k = \mathbb{E}[n_{ij}(\mathbf{x})] - \mathbb{E}[n_{ij}(\mathbf{x}|a = k)].$$

The modified objective in the offline setup needs to consider the difference for every protected label $a = k$. It is shown below:

$$\min_f \mathcal{L}(f(\mathbf{x}), y) + \lambda \sum_{i,j} \sum_k H_\delta(\mathcal{F}_{ij}^k). \tag{24}$$

To extend this to an online setting, where aggregate statistics are maintained, we need to maintain the following expectations: (a) $\mathbb{E}[n_{ij}(\mathbf{x})]$, (b) $\mathbb{E}[\nabla_\Theta n_{ij}(\mathbf{x})]$, (c) $\mathbb{E}[n_{ij}(\mathbf{x}|a = k)]$, and (d) $\mathbb{E}[\nabla_\Theta n_{ij}(\mathbf{x}|a = k)], \forall k$. This would result in an overall storage cost of $\mathcal{O}(2^h K d)$, where $d$ is the dimension of the input $\mathbf{x}$ and $K$ is the number of protected classes.

# D  Implementation Details

In this section, we describe the implementation details of the experimental setup, baselines, and training procedure.

| Dataset | Size | Split ($y$) | Split ($a$) |
|---|---|---|---|
| Adult | 32.5K | 75.9/24.1 | 66.9/33.1 |
| Census | 199.5K | 93.8/6.2 | 52.1/47.9 |
| COMPAS | 7.2K | 52.0/48.0 | 66.0/34.0 |
| CelebA | 202.6K | 85.2/14.8 | 58.3/41.7 |
| CivilComments | 33.9K | 93.4/6.6 | 76.6/23.4 |

Table 1: Dataset Statistics. We report the number of samples (size) used during online training and the percentage splits of the binary task ($y$) and protected attribute ($a$) respectively.

## D.1  Setup

We perform our experiments using the TensorFlow (Abadi et al., 2015) framework. We select the hyperparameters of the different models by performing a grid search using Weights & Biases library (Biewald, 2020). To compute the accuracy-fairness tradeoff, we run *Aranyani* on a wide range of $\lambda$'s and report the performance of all runs in the trade-off plots. We retrieve the text and image representations from Instructor and CLIP models respectively using the HuggingFace library (Wolf et al., 2020). All experiments involving *Aranyani* were optimized using an Adam (Kingma & Ba, 2015) optimizer with a learning rate of $2 \times 10^{-3}$ and Huber parameter $\delta = 0.01$. As the primary task was classification in all datasets, we use cross-entropy loss as the task loss, $\mathcal{L}(\cdot, \cdot)$. For online learning, the model is evaluated with every incoming input instance. Therefore, we perform our evaluation on the training set of each dataset, except for COMPAS and CelebA, where both the training and test set were used for online learning. We report the dataset statistics in Table 1. We report the percentage of the majority label versus the minority label (Split) in the table for all of our datasets as they have a binary task and protected label.

## D.2  Baselines

We describe the details of the baseline approaches used in our experiments.

⬭ *Aranyani* **MLP**. This is a variant of *Aranyani*, where the model $f(\mathbf{x})$ is replaced by an MLP network. We use a two-layer neural network with ReLU non-linearity to parameterize the MLP. The parameters of the MLP are: $\mathbf{W}_1 \in \mathbb{R}^{d \times m}, \mathbf{B}_1 \in \mathbb{R}^m, \mathbf{W}_2 \in \mathbb{R}^{m \times c}, \mathbf{B}_2 \in \mathbb{R}^c$, where $d$ is the input dimension, $m$ is the hidden dimension, and $c$ is the number of output class labels ($c = 1$ for regression). To compute the gradient estimates in the online setting, the following aggregate statistics are maintained: (a) $\mathbb{E}[\nabla_{\mathbf{W}_1} f(\mathbf{x}|a = k)]$, (b) $\mathbb{E}[\nabla_{\mathbf{B}_1} f(\mathbf{x}|a = k)]$, (c) $\mathbb{E}[\nabla_{\mathbf{W}_2} f(\mathbf{x}|a = k)]$, (d) $\mathbb{E}[\nabla_{\mathbf{B}_2} f(\mathbf{x}|a = k)]$, and (e) $\mathbb{E}[f(\mathbf{x}|a = k)]$ for all $k \in \{0, 1\}$. Note that the derivates are computed w.r.t. the final decisions of the MLP network $f(\mathbf{x})$, as there are no local decisions. These aggregate statistics are used to compute the gradients of the form shown below:

$$G(\Theta) = \nabla_\Theta \mathcal{L}(f(\mathbf{x}), y) + \lambda \nabla_\Theta H_\delta(\mathcal{F}), \text{ where } \mathcal{F} = \mathbb{E}[f(\mathbf{x}|a = 0)] - \mathbb{E}[f(\mathbf{x}|a = 1)].$$

In the above equation, we note that there is a single fairness term as the group fairness constraint can only be defined on the final prediction, $f(\mathbf{x})$.

◁ *Aranyani* **Leaf**. This is a similar variant of *Aranyani* as described above, where we use the proposed oblique forests and apply fairness constraints to the final prediction. Therefore, the group fairness constraint can be applied at the leaf probabilities, $p_l(\mathbf{x}) = \prod_{i=1}^h n_{i,A(i,l)}(\mathbf{x})$, as the prediction takes the following form:

$$|f(\mathbf{x}|a=0) - f(\mathbf{x}|a=1)| = \left| \sum_l (p_l(\mathbf{x}|a=0) - p_l(\mathbf{x}|a=1))\theta_l) \right|$$

$$= \left| \sum_l \left( \prod_{i=1}^{h} n_{i,A(i,l)}(\mathbf{x}|a=0) - n_{i,A(i,l)}(\mathbf{x}|a=1) \right) \theta_l \right|$$

$$\leq \sum_l \left| \prod_{i=1}^{h} n_{i,A(i,l)}(\mathbf{x}|a=0) - \prod_{i=1}^{h} n_{i,A(i,l)}(\mathbf{x}|a=1) \right| \|\theta_l\|.$$

The above expression provides an upper bound that allows us to define leaf-level fairness constraints:

$$\mathcal{F}_l = \left| \prod_{i=1}^{h} n_{i,A(i,l)}(\mathbf{x}|a=0) - \prod_{i=1}^{h} n_{i,A(i,l)}(\mathbf{x}|a=1) \right|.$$

This can be used to define the fairness gradient formulation in the online setting:

$$G(\Theta) = \nabla_\Theta \mathcal{L}(f(\mathbf{x}), y) + \lambda \sum_l \nabla_\Theta H_\delta(\mathcal{F}_l)$$

Similar to MLP gradients, the above gradients can be computed in the online setting by maintaining aggregate statistics of derivates of parameters w.r.t. the leaf-level probabilities.

■ **Hoeffding Tree-based Methods**. In general, simple ■ HT or ⬟ AHT-based baselines do not obtain good accuracy-fairness tradeoffs as they do not consider fairness at all. Note that we directly report the FAHT ▷ results for Adult and Census presented in the original paper, as we could not run the public implementation[3] and replicate the results. Since the original paper reports results only on tabular data, we could not report the results on CivilComments or CelebA. However, as HTs are not able to fit the data properly (or all) on CivilComments or CelebA datasets, incorporating additional fairness constraints would not have improved the results.

### D.3 TRAINING PROCEDURE

In this section, we provide more details about the training process presented in Section 3.4. First, we discuss the formulation of the mask used to select the node probabilities for a leaf. We have access to $\overline{\mathbf{N}} \in \mathbb{R}^{m \times 2^h}$, which stores a copy of the node decisions (as a column) for each leaf. There are $2^h$ leaves and $m = 2^h - 1$ node decisions. Using the mask $\mathbf{A}$, we wish to select the node decisions needed to compute each leaf probability. For example, consider the mask for a tree with height 2, which has 4 leaves (number of columns) and 3 internal nodes (number of rows):

$$\mathbf{A} = \begin{bmatrix} 1 & 1 & -1 & -1 \\ 1 & -1 & 0 & 0 \\ 0 & 0 & 1 & -1 \end{bmatrix}$$

Now let us consider the probability of reaching the second leaf of the tree involves node decisions of the root (index 0) and leftmost node of the first level (index 1). The *highlighted* column in the above equation selects the desired nodes. Note the mask entry takes the value 1 when the leaf can be reached by choosing the left path from the node, $-1$ when the leaf is reachable from the right, and 0 when the leaf is unreachable from the node. For trees with different height $h$, the entries of $\mathbf{A}$ can be derived using the following general form:

$$\mathbf{A}_{ij} = \begin{cases} 1, & \text{if } 2^{(h-l)}(i+1) \leq 2^h + j < 2^{(h-l)}(i+1) + 2^{(h-l-1)} \\ -1, & \text{if } 2^{(h-l)}(i+1) + 2^{(h-l-1)} \leq 2^h + j < 2^{(h-l)}(i+1) + 2^{(h-l)} \\ 0, & \text{otherwise.} \end{cases}$$

---

[3]https://github.com/vanbanTruong/FAHT/

---

**Algorithm 1** *Aranyani* Online Learning Algorithm

---

1: **Input**: Oblique tree with parameters $\mathbf{W}, \mathbf{B}, \Theta$.
2: **for** $a \in \{0, 1\}$ **do**
3:     $c_a = 0$ // set the sample count for label $a$
4:     // aggregate node outputs and their gradients for label $a$
5:     $\mathbf{N}_a = \mathbf{0}_m, \nabla_{\mathbf{W}} \mathbf{N}_a = \mathbf{0}_{m \times d}, \nabla_{\mathbf{B}} \mathbf{N}_a = \mathbf{0}_m$ // where $m = 2^{h-1}$
6: **end for**
7: // Begin online learning
8: **for** $\mathbf{x}_t \in \mathcal{X}$ **do**
9:     // Get the prediction as described in Section 3.4.
10:     $\mathbf{N} = g(\mathbf{W}^T \mathbf{x}_t + \mathbf{B})$
11:     $\hat{y} = \exp\left(\mathbf{1}_{1 \times m} \log \mathbf{P}\right)\Theta$
12:     Access true labels $(y, a)$ after prediction
13:     $c_a = c_a + 1$ // Update the counts
14:     // Update the aggregate statistics
15:     $\mathbf{N}_a = \mathbf{N}_a(1 - 1/c_a) + \mathbf{N}/c_a$
16:     $\nabla_{\mathbf{W}} \mathbf{N}_a = \nabla_W \mathbf{N}_a(1 - 1/c_a) + \nabla_{\mathbf{W}} \mathbf{N}/c_a$
17:     $\nabla_{\mathbf{B}} \mathbf{N}_a = \nabla_B \mathbf{N}_a(1 - 1/c_a) + \nabla_{\mathbf{B}} \mathbf{N}/c_a$
18:     // Update the aggregate statistics
19:     $\hat{\mathcal{F}} = \mathbf{N}_0 - \mathbf{N}_1 \in \mathbb{R}^m$
20:     $\nabla_{\mathbf{W}} \hat{\mathcal{F}}_{\mathbf{W}} = \nabla_{\mathbf{W}} \mathbf{N}_0 - \nabla_{\mathbf{W}} \mathbf{N}_1 \in \mathbb{R}^{m \times d}$
21:     $\nabla_{\mathbf{B}} \hat{\mathcal{F}}_{\mathbf{B}} = \nabla_{\mathbf{B}} \mathbf{N}_0 - \nabla_{\mathbf{B}} \mathbf{N}_1 \in \mathbb{R}^m$
22:     // Update the tree parameters using gradients from Equation 5
23:     $\mathbf{W} = \mathbf{W} - \eta \left[ \nabla_{\mathbf{W}} \mathcal{L}(\hat{y}, y) + \lambda \nabla_{\mathbf{W}} H_\delta(\hat{\mathcal{F}}) \right]$
24:     $\mathbf{B} = \mathbf{B} - \eta \left[ \nabla_{\mathbf{B}} \mathcal{L}(\hat{y}, y) + \lambda \nabla_{\mathbf{B}} H_\delta(\hat{\mathcal{F}}) \right]$
25:     $\Theta = \Theta - \eta \nabla_\Theta \mathcal{L}(\hat{y}, y)$
26: **end for**

---

where $l = \lfloor \log_2(i + 1) \rfloor$. Using the above mask $\mathbf{A}$, we can compute $f(\mathbf{x})$ efficiently and train it using autograd libraries via backpropagation.

We also provide an outline of *Aranyani*'s online training algorithm in Algorithm 1. This algorithm showcases how *Aranyani* can be trained efficiently. The fairness gradients for all nodes are computed simultaneously using matrices of aggregate statistics $(\mathbf{N}_a, \nabla \mathbf{N}_a)$. The task gradients $\nabla \mathcal{L}(\cdot, \cdot)$ are efficiently using standard autograd libraries.

# E  ADDITIONAL EXPERIMENTS

In this section, we provide the details of additional experiments we perform to analyze the performance of *Aranyani*.

**Ablations with $\lambda$.** In this experiment, we perform ablations by varying the $\lambda$ parameter (Equation 5), which allows us to control the accuracy-fairness trade-off. In Figure 5, we report the accuracy and DP scores on the Adult dataset during online learning. We observe that increasing $\lambda$ results in lower accuracy and improved DP consistently throughout the training process. This shows that *Aranyani* presents a general framework that allows the user to control accuracy-fairness trade-offs using $\lambda$.

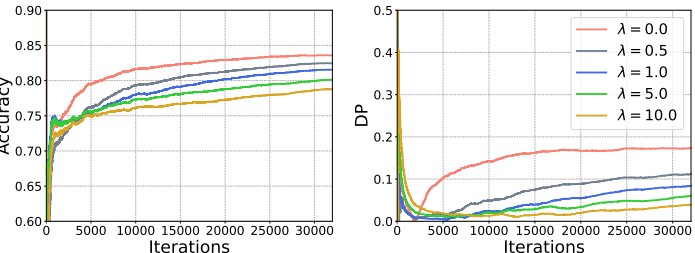

Figure 5: Ablations with different $\lambda$. We observe that increasing $\lambda$ results in lower accuracy and improved DP scores consistently throughout the online learning process.

**Tree Ablations.** In this experiment, we perform ablation experiments to investigate the impact of the number of trees in the oblique forest on the accuracy-fairness trade-off. In Figure 6, we report the change in average accuracy and demographic parity achieved during the online learning process in Adult dataset, when the number of trees in *Aranyani* is increased. In this experiment, we set the hyperparameter $\lambda = 0.1$. We observe a gradual decrease in accuracy (Figure 6 (left)) when the number of trees is increased, which can potentially result from overfitting. In a similar trend, we observe an improvement in the demographic parity (Figure 6 (right)), which is caused by the drop in accuracy due to overfitting. We report the results over 5 different runs for each setting. The error bars in the plots illustrate the standard deviation within each of the settings. We observe that the standard deviation (in Accuracy and DP) gradually decreases with an increased tree count.

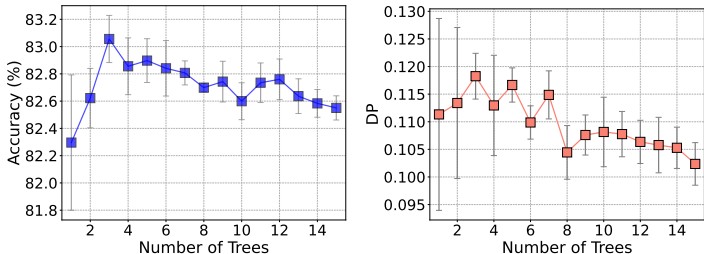

Figure 6: Ablation experiment with a varying number of trees, $|\mathcal{T}|$, in the oblique forest of *Aranyani*. We observe a slight drop in accuracy and a consistent drop in demographic parity when the number of trees used in *Aranyani* is increased.

**Gradient Convergence.** In this experiment, we investigate the convergence of fairness gradients (derived in Equation 5). We perform experiments on CivilComments dataset and report the gradient norms for all node parameters $(\mathbf{W}, \mathbf{B})$ in Figure 7 (left & center). The $y$-axis in the figure is in log-scale for better visibility. We observe that the fairness gradients for both parameters converge over time and it is well correlated with the demographic parity of the decisions during online learning.

In Figure 7 (right), we study the gradient convergence bounds predicted by Theorem 2. Specifically, we try to understand how the fairness gradient norm varies as a function of the tree height. We observe a linear correlation between the magnitude of the fairness gradient norm and the height of the tree. However, in general, for small tree heights ($h \leq 10$), we observe that the gradient bound is quite slow and doesn't impact the final demographic parity scores (which lie between $\sim 0.05$-$0.07$).

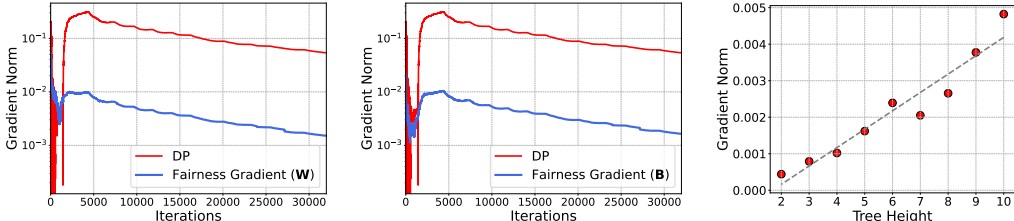

Figure 7: Convergence of gradients on CivilComments dataset: (left & center) We show the evolution of the gradient norms of the oblique tree parameters ($\mathbf{W}, \mathbf{B}$) and the demographic parity during the online training process. We observe that the fairness gradients converge along with demographic parity. (right) We report the norm of the fairness gradients (for $\mathbf{W}$) at the end of online training with different tree heights. We observe that the gradient magnitude is very small and there appears to be a linear correlation with tree height.

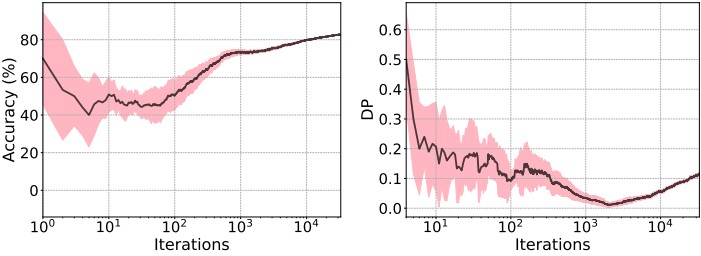

Figure 8: Variation in the Accuracy and DP scores during online learning using *Aranyani* on the Adult dataset. The $x$-axis is shown in $\log$-scale. We observe that most of the variance is concentrated in the initial parts of the training process.

**Performance Variance**. In this experiment, we investigate the variance in performance during the training process. We perform online learning using *Aranyani* on Adult dataset using 5 different seeds. In Figure 8, we report the average Accuracy and DP scores during online learning. The standard deviation for each metric is highlighted in light red. Note that the $x$-axis is shown in log scale to observe the variance in performance clearly. We observe that most of the variation in metric is concentrated in the initial parts of the training process. We observe a minimal variance in the scores after a small number of around 1000 iterations. This showcases the robustness of *Aranyani* during the online learning process.

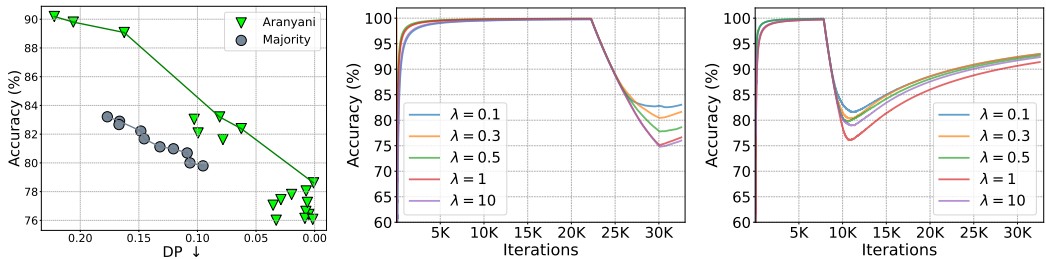

Figure 9: Performance of *Aranyani* with an adversarial stream of data on the Adult dataset. Left: We report the Accuracy vs. DP tradeoff for Adult dataset with an adversarial stream. Right: We report the accuracy of *Aranyani* during the online learning process for different $\lambda$. We observe a significant dip in accuracy when the minority class is introduced.

**Adversarial Stream**. In this experiment, we investigate the robustness of *Aranyani* in the advent of an adversarial stream of samples. We experiment on the Adult dataset and construct an adversarial online stream in two different ways: (a) In the first stream, we present *Aranyani* with 90% of the majority class samples, followed by all samples of the minority class, and then the remaining 10% of the majority class samples. In Figure 9 (left), we report accuracy vs DP tradeoff and observe that

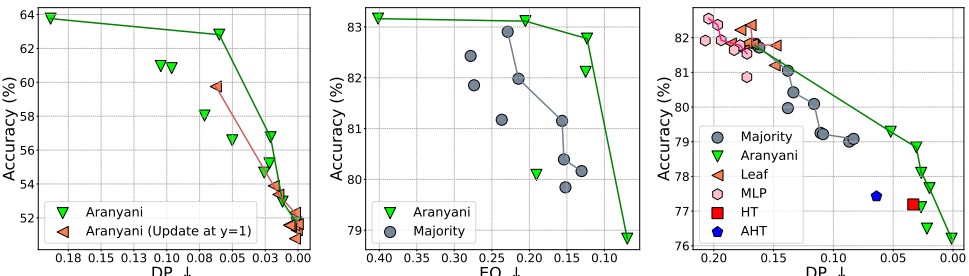

Figure 10: (Left) Performance of *Aranyani* under limited feedback where the model is updated only when the prediction $\hat{y} = 1$. (Center) Performance of *Aranyani* under the equality of opportunity fairness constraint. (Right) Performance of *Aranyani* in the batch setting where 10 instances are encountered at each time step.

*Aranyani* performs significantly better than the strong majority baseline. In Figure 9 (Center), we report the accuracy achieved by *Aranyani* during the online learning process. We observe a sharp dip in accuracy when the minority class is introduced, which is expected. The accuracy improves towards the end and results using different $\lambda$ values show that the accuracy can be controlled using it. (b) In the second stream, we follow the same procedure as in the first one but replace the majority class samples with the minority ones. In Figure 9 (Right), we observe a similar dip in accuracy when the majority class is introduced but the performance gradually improves over time.

Overall, this experiment shows that *Aranyani* is susceptible to adversarial data streams like any other ML system. However, our results show that even in such cases the user can control the fairness-accuracy tradeoff.

**Limited Feedback**. In this experiment, we explore scenarios where *Aranyani* conditionally receives feedback. Specifically, during online training *Aranyani* receives feedback from the environment only when its prediction, $\hat{y} = 1$. In Figure 10 (left), we report the performance of *Aranyani* on the COMPAS dataset. We observe that *Aranyani* achieves similar tradeoff curves but it is unable to achieve high accuracies for a similar set of $\lambda$'s compared to when full feedback is provided. This is expected as the system is unable to get feedback for many of the input samples.

**Equality of Opportunity**. In this experiment, we evaluate the efficacy of *Aranyani* for the fairness notion of equality of opportunity (EO). For equality of opportunity, the node-level constraint is:

$$\mathcal{F}_{ij}^c = \mathbb{E}[n_{ij}(\mathbf{x}|y = 1, a = 0)] - \mathbb{E}[n_{ij}(\mathbf{x}|y = 1, a = 1)].$$

We report the accuracy vs EO results in Figure 10 (center). We observe that *Aranyani* achieves a much better trade-off than the strong majority baseline. This showcases the efficacy of *Aranyani* while using different fairness measures.

**Batch Learning**. In this experiment, we analyze the performance of *Aranyani* in the batch setting where we present the system with 10 input instances at each time step. We report the results in Figure 10 (right). We observe trends similar to the online learning setting with *Aranyani* achieving the best fairness-utility tradeoffs. This shows the efficacy of *Aranyani* even when input instances are introduced in a batch-wise manner.

**Runtime Analysis**. We empirically analyze the runtime of *Aranyani* and other baseline approaches on the Adult dataset. We report the total time to process an input stream of 32K samples in Table 2. We observe that Ho-effding tree-based approaches are the fastest as they do not require any gradient propagation. Among the variants of *Aranyani*, *Aranyani* (MLP) achieves the fastest runtime as it requires fairness gradient computation only w.r.t. the final network output.

| Method | Runtime (min) |
| --- | --- |
| HT | 5.07 |
| AHT | 5.67 |
| *Aranyani* | 32.97 |
| *Aranyani* (MLP) | 14.65 |
| *Aranyani* (Leaf) | 37.12 |

Table 2: Runtime of different online fairness mitigation approaches.

**Regularizer Ablation**. We investigate the performance of *Aranyani* with regularizers different from the Huber function. Specifically, we focus on the L2 norm as a regularizer. For using L2-norm in

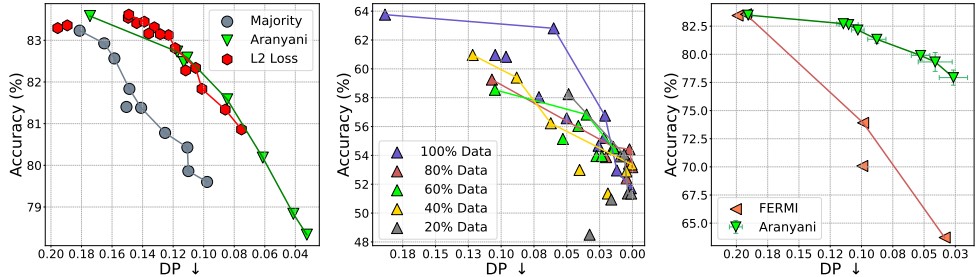

Figure 11: (Left) Performance of *Aranyani* using a L2 regularization for the node constraints $\mathcal{F}_{ij}$ in Adult. (Center) Ablation experiment to investigate the performance of *Aranyani* under a limited data regime on the COMPAS dataset. (Right) Comparison of *Aranyani* with stochastic batch fair learning technique, FERMI.

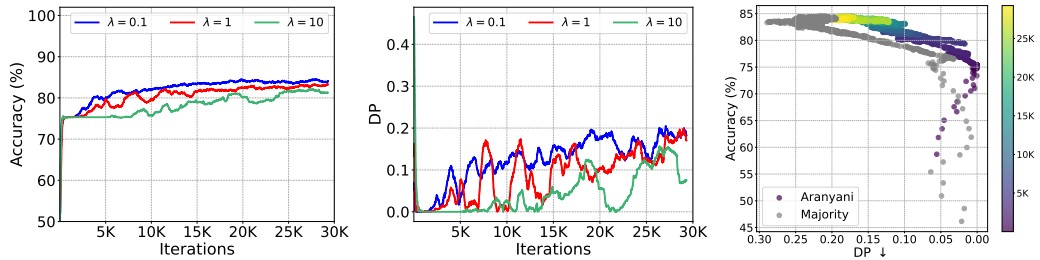

Figure 12: (Left & Center) Evaluation of *Aranyani* on an unseen held-out validation set during online learning. We observe that the accuracy score improves consistently while there is variance in the DP scores. (Right) Convergence of the tradeoff curve over training iterations (shown in colors) during online learning.

the online setup, the fairness gradient can be written as $\nabla_{\Theta} L_2(\mathcal{F}_{ij}) = \mathcal{F}_{ij} \nabla_{\Theta} \mathcal{F}_{ij}$. In Figure 11 (left), we report the results of this setup on the Adult dataset. We observe that *Aranyani* using L2 regularization achieves similar fairness-accuracy tradeoffs. However, it is unable to reduce the DP scores beyond a certain extent. We hypothesize that this phenomenon happens as the approximation error for L2 gradients is large, which eventually hinders the convergence of fairness gradients.

**Dataset Size Ablation**. In this experiment, we aim to investigate the impact on the performance of *Aranyani* when exposed to varying fractions of data during online learning. In Figure 11 (center), we report the results of this experiment on the COMPAS dataset. We observe that the fairness-accuracy tradeoff gradually deteriorates with a decreasing amount of data. In particular, we note that *Aranyani* fails to attain higher accuracies, as anticipated due to the reduced size of the training data.

**Comparison with Batch Techniques**. In this experiment, we compare *Aranyani*'s performance with the stochastic batch-based fair learning method, FERMI (Lowy et al., 2022), which is the only fairness algorithm we are aware of that can be applied with a batch size of 1. In Figure 11, we report the fairness-accuracy tradeoff for *Aranyani* and FERMI. We observe that *Aranyani* achieves a much better fairness-accuracy tradeoff than FERMI in the online setting with a batch size of 1. *Aranyani* can consistently beat FERMI across different values of $\lambda$. Contemporary work (Baharlouei et al., 2024) proposed algorithms to make stochastic versions of offline algorithms amenable to small batch sizes. Future works can investigate the performance of such algorithms in online settings.

**Temporal Analysis**. Inspired by the notion of fairness in hindsight (Gupta & Kamble, 2021), we try to evaluate how the fairness of *Aranyani* varies over time. In the online setting (described in Section 3.1), the system is evaluated on each incoming new sample. To evaluate the fairness of *Aranyani* in a more absolute setting, we select 10% of Adult's data as a held-out validation set. We measure the fairness (DP) and utility (Accuracy) over time. In Figure 12 (left & center), we observe a gradual improvement in accuracy scores over time and there is a slightly higher variance in the DP scores. However, we find that there it is still possible to control the fairness-utility tradeoff using $\lambda$.

**Tradeoff Convergence**. In this experiment, we investigate the convergence of the fairness-utility tradeoff achieved by *Aranyani* during the online learning process. Specifically, we plot the DP and accuracy scores on a held-out validation set attained by *Aranyani* at each iteration. In Figure 12 (right) we observe that gradually improves over time and the performance leans slightly more towards the accuracy at the end. The color of each point indicates the iteration when that tradeoff was achieved. We also plot the convergence curve for the Majority baseline and observe that *Aranyani* consistently outperforms it during the training process.

