# OpenReview forum: "Enhancing Group Fairness in Online Settings Using Oblique Decision Forests"
_ICLR.cc/2024/Conference — ICLR 2024 spotlight_

### Official Review · Reviewer_uVjM · 2023-10-22

**Soundness:** 4 excellent
**Presentation:** 4 excellent
**Contribution:** 3 good
**Rating:** 6
**Confidence:** 4

**Summary:**

This study introduces an online algorithm called Aranyani, aimed at addressing group fairness issues. This algorithm integrates a collection of oblique decision trees and leverages the tree's structural attributes to improve group fairness in online learning scenarios.

Firstly, the study demonstrates that when group fairness constraints are applied to decisions at the local node level, it results in parameter isolation, paving the way for superior and more equitable outcomes.

Secondly, by keeping track of the overall statistics of decisions made at the local node level, the algorithm can effectively calculate group fairness gradients, removing the requirement for extra storage for forward/backward passes.

Thirdly, the study offers a framework work for training Aranyani.

Lastly, both empirical and theoretical evidence provided in the study underscores the efficacy of the suggested algorithm.

**Strengths:**

* This study focuses on a unique and critical problem wherein machine learning fairness is examined in an online context, with individual sample instances observed sequentially.
* In such a scenario, determining group fairness attributes and executing backward propagation to modify model parameters necessitates retaining all prior samples, leading to computational complexities. To address this, the paper introduces an effective framework for forward and backward propagations, taking advantage of statistical summaries from previous samples.
* The presented algorithm consistently surpasses baseline techniques across all tasks. It is both empirically and theoretically robust and comprehensive.

**Weaknesses:**

* Utilizing the aggregate of past statistics to adjust the model parameters poses a challenge. As the model parameters shift with each time step, past statistics, derived from earlier model parameters, might diverge significantly from what would be obtained if the model gradients were calculated using all the previously stored data. I'm curious if the authors could elaborate on the conditions under which using aggregated past gradients to update the current model would be successful or not.
* The problem is framed in an online setting where the actual predictive labels are seen after the model makes its predictions. More realistically, if the model consistently underperforms, it could result in a shift in data distribution. Specifically, the minority group might cease supplying data for model updates. Could the author discuss how the proposed algorithm would operate under these circumstances?

**Questions:**

Questions are provided in the weakness section.

---

> ### Author Response · Authors · 2023-11-18
>
> Thank you for your detailed feedback and helpful comments.
>
> > Conditions for online learning using stale gradients
>
> Our proposed method, Aranyani, is a tree structure specialized and related learning algorithm designed to effectively use such aggregated computations. The tree structure is parameterized using a logistic network at each node. We compute fairness gradients by taking the derivative of node outputs. Crucially, this structure allows us to prove that the error in computing the fairness gradients cannot diverge in an unbounded manner.
>
> The theoretical results we propose rely solely on the assumptions of bounded inputs, a compact parameter set, and smooth activation functions, as well as a smooth task loss. This allows us to derive the explicit bound in Lemma 3 (Page 6), which shows that the error is a function of the $\delta$ parameter (associated with the Huber loss) and input bound $B$.
>
> > Aranyani's performance under an adversarial input stream.
>
> Indeed understanding the performance of Aranyani under shifts in data distribution is an interesting research question. We have added results in an adversarial setting that is aligned with your proposal and report them in Appendix D (Subsection: Adversarial Stream).
>
> Specifically, we construct an adversarial stream of data using the Adult dataset, where we present Aranyani with 90% of the majority class samples followed by the entire minority class and the remaining 10% of the majority class samples. We also experiment with the inverse scenario where the minority class samples are presented before the majority ones.
>
> The detailed results are presented in Appendix D (Subsection: Adversarial Stream) section. From the results, we observe that even in such scenarios Aranyani can achieve good fairness-utility tradeoffs and it is possible to control the tradeoff using the $\lambda$ parameter.
>
> We would also like to note that the convergence bounds in Theorem 1 are independent of the nature of the data stream. Therefore, the gradient bounds of Aranyani always converge to a small value, emphasizing good performance.

---

> ### Author Response · Authors · 2023-11-21
>
> Thanks for taking the time to review our responses. Please let us know if you have any questions or would like any further clarifications.

---

### Official Review · Reviewer_VSWz · 2023-10-31

**Soundness:** 2 fair
**Presentation:** 2 fair
**Contribution:** 3 good
**Rating:** 5
**Confidence:** 3

**Summary:**

This paper proposes a method for achieving group fairness in online setting, equipped with fairness, accuracy, convergence analysis. Moreover, numerical experiments have been conducted for showing the great performance of the proposed method in terms of reducing the unfairness while maintaining the accuracy.

**Strengths:**

1. This paper proposed an effective approach called Aranyani, and provide solid theoretical analysis for it.
2. The experiment results look promising

**Weaknesses:**

1. [Major] The main use cases for achieving fairness in this online setting are missing. This issue makes the motivation of this work lack justification.
2. [Major] Problem setting should be stated more clearly. For instance, in this paper, considers an online setting where only one single instance can be processed. Typical online learning indeed process one data point at a time but also privy to the feedback from previous data points, but this paper seems to assume that the previous data points are not available. A more clearly setup should be stated before getting into the gradients used in the algorithm (5).
3. [Major] Authors do not clearly state the goal in the online setting. Can you add an equation like (3) to the online setting? I am not even sure what is the right fairness metric for online settings — at each step, are you measuring the fairness for the entire dataset?

**Questions:**

1. Where is the FAHT’s result in the rightmost plot in Figure 2?
2. Why is FAHT worse than the adaptive HT in terms of DP?

---

> ### Author Response · Authors · 2023-11-18
>
> Thank you for your detailed feedback and helpful comments.
>
> > Use cases of group fairness in online settings.
>
> Studying group fairness in the online setting has several use cases (outlined below). In our initial draft, we alluded to these in the last sentence of the first paragraph of the introduction. Based on your feedback, we have updated the introduction in the revised draft to more clearly elaborate on these use cases.
>
> One major use case for improving fairness in online settings is applications that involve output moderation using safety/toxicity classifiers), which may include chatbots, comments on social media platforms, etc. In such applications, the definition of safety is evolving and new unsafe data points are identified on the fly, which makes them a prime candidate for online learning. Our goal is to improve demographic fairness for models learned through online learning in these applications.
>
> To showcase the practical feasibility of our approach, we conducted experiments aimed at detecting toxicity in Wikipedia comments (refer to Section 5.2) within an online environment as a simulation testbed for this problem.
>
> In a similar manner, we have used other datasets to simulate online learning settings, where data would arrive continuously over time, e.g., CelebA, an—image classification dataset. The task for this dataset is to predict certain characteristics of images (in this case hair color). One can imagine that fairness in predicting image characteristics would be crucial to services that organize user photos. Such image characteristics may be used for organization; fairness ensures consistent organization experience for users.
>
> > Fairness-aware online learning setting and the significance of storing past samples within this context.
>
> As mentioned by the reviewer, to achieve group fairness in online settings, we assume that we can only store aggregate feedback from previous data points (e.g., aggregate of previous gradients). We assume that we cannot store the actual data points that contain sensitive demographic information.
>
> Unlike typical online learning setups, it is not possible to obtain a fairness regularizer gradients update using a single instance. This is because group fairness regularizers are defined using a group of instances. This poses an interesting challenge of updating the model at each step without accessing previous instances.
>
> This is the major challenge that we overcome in this paper. We have updated Section 3.1 to articulate the setup and our contribution better.
>
> > Objectives and metrics in group fairness aware online learning.
>
> The online learning setup in our case is to perform multi-objective optimization for task loss and group fairness loss.
>
> Since the fairness loss is defined over groups of samples it is not possible to define the fairness loss for a single instance and compute a instance-level loss. This makes it difficult to apply conventional online learning algorithms to this problem.
>
> In principle, we care about the fairness-accuracy trade-off achieved by the system’s predictions over time. Therefore, in the online setup, Eq. 2 can be written as:
>
> $\min_f \sum_{t=1}^T \mathcal{L}(f_t(\mathbf x), y), \text{ subject to DP = } \left|\mathbb{E}[f_t(\mathbf x|a=0)] - \mathbb{E}[f_t(\mathbf x|a=1)]\right| \leq \epsilon.$
>
> Please note that the expectations in the above equation are over samples that have not appeared in the online training stream. This equation can be easily modified into Aranyani's objective by replacing the DP constraint with the node-level constraints in the tree (mimicking the transition from Eq. 2 $\rightarrow$ Eq. 3).
>
> In online learning, we can typically get an unbiased estimate of this by computing objectives (training loss) on the new sample before updating the model. However, due to our unique problem setup, the group fairness objective cannot be computed using a single instance.
>
> We empirically investigate the performance of our model by computing the fairness-accuracy tradeoff and observe how it evolves during the online training process. We report the results in Appendix D, Subsection: Tradeoff Convergence, Figure 12 (right). We observe that Aranyani achieves gains over the majority baseline, which implies that it can achieve non-trivial tradeoff gains.
>
> **Evaluation**: To evaluate this in practice, we store all predictions made by our system and report the group fairness (demographic parity) using these decisions at each time step.

---

> > ### Author Response · Authors · 2023-11-18
> >
> > > Missing FAHT results on COMPAS
> >
> > In the updated draft, we have incorporated the results of FAHT for the COMPAS dataset (rightmost plot of Figure 2).
> >
> > Before submission, we had difficulty re-running and reproducing the Java-based FAHT implementation on our own and reported the FAHT results directly from their paper (on the datasets the authors evaluated: Adult & Census). With additional work, we were able to modify the FAHT codebase to run on the remaining datasets. Updated results are reported in Figure 2. We observe that Aranyani consistently outperformed FAHT. For example, on the Adult dataset, Aranyani can significantly reduce the DP score (0.19 $\rightarrow$ 0.11) while achieving similar levels of accuracy.
> >
> > > Comparison of the performance between FAHT and adaptive HT
> >
> > We have replicated the FAHT baseline and updated the results in Figure 2. The updated results show that FAHT slightly lags behind adaptive HT in DP only in the Adult dataset. The underlying implementations of Hoeffding Trees differ between FAHT and adaptive HT. FAHT is implemented in Weka ([FAHT github](https://github.com/vanbanTruong/FAHT)) and Adaptive HT in sklearn ([Adaptive HT github](https://github.com/scikit-multiflow/scikit-multiflow/blob/a7e316d1cc79988a6df40da35312e00f6c4eabb2/src/skmultiflow/trees/hoeffding_adaptive_tree.py)). The scikit-learn implementation of Adaptive HT exhibited numerical instabilities at times that could have affected its convergence. We hypothesize that this issue might have influenced the results on the Adult dataset.

---

> > > ### Author Response · Authors · 2023-11-21
> > >
> > > Thanks for taking the time to review our responses. Please let us know if you have any questions or would like any further clarifications.

---

> > > > ### Comment · Reviewer_VSWz · 2023-11-22
> > > > **Thanks for your detailed comments.**
> > > >
> > > > Thanks for your detailed comments. The responses have cleared up most of my concerns. I still have a few questions
> > > >
> > > > 1. I am a bit confused by the optimization problem for the online setting — why do we just consider achieving DP for t? Or do you mean we need to achieve this constraint for all t?
> > > > 2. Thanks for clarifying the specific setting you are considering. I am still not convinced why it is not possible to obtain a fairness regularizer gradients update using a single instance. For instance, if you look at equation (5) in the paper “A Reductions Approach to Fair Classification”, the regularizer can be just represented as some linear combination of losses → therefore, we can still compute the gradient just like typical online learning setting, and then use the FTL, right? Please correct me if I am wrong. In fact, representing the fair learning objective as the linear combination of losses is pretty common, not specific to this paper I just pointed out. This paper also does the same thing: “FairBatch: batch selection for model fairness”.
> > > >
> > > > While this paper is considered a novel setting, I agree that this paper can be a good starting point. However, I am still not fully convinced of the importance of the motivation. Therefore, I am willing to increase my score to 5, and I look forward to further discussion.

---

> ### Author Response · Authors · 2023-11-22
> **Response to follow-up question from Reviewer VSWz**
>
> First, we would like to express our gratitude to the reviewer for taking the time to read our response, engaging with it, increasing the score, and allowing us to engage in an interactive discussion to address the remaining comments. Below, we respond to the two specific remaining comments.
>
> > *Why do we just consider achieving DP for t? Or do you mean we need to achieve this constraint for all t?*
>
> Yes, you are correct. We mean that the DP condition needs to be satisfied for all t. Here is the updated objective for the same:
>
> $\min_f \sum_{t=1}^T \mathcal{L}(f_t(\mathbf x), y), \text{ subject to DP = } \left|\mathbb{E}[f_t(\mathbf x|a=0)] - \mathbb{E}[f_t(\mathbf x|a=1)]\right| \leq \epsilon. \forall t \in [1, T]$
>
>
> > ...... [As in] Eq. (5) in the paper “A Reductions Approach to Fair Classification”, the regularizer can be just represented as some linear combination of losses → therefore, we can still compute the gradient just like typical online learning setting, and then use the FTL, right? [...] [Similar approach in] “FairBatch: batch selection for model fairness”*
>
>
> Thanks for pointing this out.
>
> Equation 5 (below) in (Agarwal et al. 2018 [1]) is the definition of cost sensitive classification. ~describes the task loss and not the fairness regularizer.~
>
> $$\arg \min_{h \in \mathcal H} \sum^n_{i=1} h(X_i) C^1_i + (1 - h(X_i)) C^0_i. \quad \text{(5)}$$
>
> ~In Agarwal et al’s Eq (5), each class has an associated cost ($C_i^0$ for class 0, $C_i^0$ for 1). This task loss (Eq 5) is additive across input samples. Similarly in our paper the task loss (our Eq 2), is additive across input samples.~
> The fairness regularizers in Agarwal et al are reformulated as cost-sensitive classification. However, their cost-sensitive optimization procedure requires estimation of certain probability distributions for it to work well as we will discuss below:
>
> The challenge for online methods comes in the fairness regularizer in the objectives. Agarwal et al. (2018)  applies the fairness regularizer in the first (unlabeled) equation of Section 3.2 in their paper shown below.
>
> $$L(Q, \mathbf{\lambda}) = \widehat{\mathrm{err}} (Q) +  \mathbf{\lambda}^T (\mathbf M \mathbf{\hat{\mu}}(Q) - \mathbf{\hat{c}}).$$
>
> In this equation, $\widehat{\mathrm{err}} (Q)$ refers to classification error, and the fairness objective is $\mathbf{\hat{\mu}}(Q)$ term is a vector of conditional moments across protected groups ($a$) or target label ($y$) or both (described in Example 1&2 in their paper). This is shown below:
>
> $$\widehat{\mathbf{\mu}}(Q) = [\widehat{\mu}_{a_i, y_i}(Q)], \quad \forall (a_i, y_i)$$
>
> Recall that the setting in our paper considers a single sample at a time. This sample will belong to one of the two protected groups (the attribute $a$). Let’s consider DP fairness in such a setting, $\hat{\mu}(Q) = [\hat{\mu_{a=0}}(Q), \hat{\mu_{a=1}}(Q)]$
>  (since there is no conditioning on the true label). Suppose the observed sample has $a=0$. We would then not be able to estimate the fairness term, $\hat{\mu_{a=1}}$. We can estimate $\hat{\mu}_{a=0}$,  but note that the overall estimate of the cost-sensitive objective function could be severely biased if we simply plug in $\hat{\mu}$ (and hence the overall optimization procedure may not even converge) ~but the estimate could be severely biased (and hence not converge).~
>
> Similarly, in the (Roh et al., 2021 [2]) paper the additive equations are presented in Section 2 using $L_{y, z}$ terms shown below:
>
> $$\mathbf{w_\lambda} = \arg\min\limits_{\mathbf{w}} \lambda_1 L_{0,0}(\mathbf{w}) + \left(\frac{m_{*,0}}{m} - \lambda_1\right)L_{1,0}(\mathbf{w}) + \lambda_2 L_{0,1}(\mathbf{w}) + \left( \frac{m_{\*, 1}}{m} - \lambda_2\right) L_{1,1}(\mathbf{w}).$$
>
> Quoting from the notation section of their paper: `` $L_{y,z}(\mathbf w)$ be the empirical risk aggregated over samples subject to $y = y$ and $z = z$”. Since a given input sample is associated with only a single $y’$ and $z’$, all terms except the $L_{y’, z’}(\mathbf w)$ cannot  be estimated using a single sample.
>
> In summary, the difficulty of group fairness in the online setting is about ~estimating~ deriving an unbiased estimator for the fairness ~term~ regularizer from a single observed sample. This is evident in both of the approaches mentioned, Agarwal et al (2018) and Roh et al (2021). Concretely, if we have a binary classifier with a binary sensitive attribute $a$. With a single sample, with say $a=0$, it is not clear how to estimate the fairness violation rate from this single sample since the other attribute $a=1$ is unobserved.
>
>
>
> [1] A Reductions Approach to Fair Classification, Agarwal et al. 2018 [(link)](https://arxiv.org/pdf/1803.02453.pdf)
>
> [2] FairBatch: Batch Selection For Model Fairness, Roh et al. 2021 [(link)](https://arxiv.org/abs/2012.01696)
>
> We hope that this helps clarify your question. Please don’t hesitate to let us know if we can help answer any other questions/comments in the remainder of the discussion period.

---

> > ### Comment · Reviewer_VSWz · 2023-11-22
> > **Thanks for the response.**
> >
> > > Equation 5 (below) in (Agarwal et al. 2018 [1]) describes the task loss and not the fairness regularizer.
> >
> > No, it is indeed describing the fair loss. It formulates a fairness-regularized objective and then shows that it is equivalent to equation 5 -> please see example 3 and 4 in their paper.

---

> ### Author Response · Authors · 2023-11-23
> **Thanks for your follow-up; clarifying our previous response**
>
> We apologize for the miscommunication on our end; Eq. (5) is indeed the definition of vanilla cost sensitive optimization (and neither the task loss or the overall loss). We have corrected this in our above reply through an edit for clarity as well.
>
> Apart from this, we should say that the main point of our response is not dependent on Eq. (5). The main point that we are making is that the reduction approach of Agarwal et al (2018) relies on the computation of the estimator $\hat{\mu}$, where $\mu$ is an expectation over an indicator function as defined in the unnumbered Eq. after (1). This is the term that cannot be directly estimated from a single (or even no) sample in our online learning setting. Thus, enforcing the constraints on these quantities is not directly possible and is not expected to work.
>
> Suppose the observed sample has $a=0$. We would then not be able to estimate the fairness term, $\hat{\mu_{a=1}}$ as we have no samples on it. Estimating $\hat{\mu}$ is needed in the learning algorithm for the cost-sensitive reduction.

---

### Official Review · Reviewer_mq1x · 2023-10-31

**Soundness:** 4 excellent
**Presentation:** 4 excellent
**Contribution:** 4 excellent
**Rating:** 8
**Confidence:** 4

**Summary:**

This paper introduces "Aranyani," a novel framework designed to compute and enhance group fairness in online learning scenarios. The work is systematically presented, starting with foundational knowledge on oblique decision trees in Chapter 2. Chapter 3 delves into the problem formulation, detailing demographic parity equations for both offline and online settings. Within this chapter, the authors explain their choice of the Huber loss function, addressing the challenges inherent to online scenarios. Chapter 4 provides a comprehensive theoretical analysis of Aranyani, covering its expectations and assumptions. The experimental findings, presented in Chapter 5, benchmark Aranyani against prominent online learning algorithms.

Central to the paper is Aranyani's innovative approach to achieving fairness in online settings. Aranyani leverages an ensemble of oblique decision trees, which are capable of making nuanced oblique splits by utilizing routing functions that account for all input features. The authors begin by detailing an offline setting, constraining the objective function with a fairness objective (L1 norm), and then adapt this framework to oblique decision forests. Given the non-smooth and convex nature of the L1 norm, optimization challenges arise. The authors cleverly employ Huber Regularization to smoothen the L1 norm, paving the way for more effective optimization.

What sets Aranyani apart is its approach to data storage. Traditional methods tend to store instances to train a model, whereas Aranyani calculates fairness gradients based on aggregated statistics from previous decisions, effectively handling the online situation.

From a theoretical standpoint, the authors rigorously examine four fundamental properties of Aranyani: Demographic Parity (DP), Rademacher Complexity, Fairness Gradient Estimation, and Gradient Norm Convergence. Each of these properties is supported by robust theoretical proofs.

To validate their approach, the authors conduct experiments on five diverse datasets: UCI Adult, Census, COMPAS, CelebA, and CivilComments. Their results indicate that, in terms of the Fair-Accuracy trade-off, Aranyani outperforms existing online learning algorithms across most datasets. Notably, its performance on the COMPAS dataset was less dominant, warranting further investigation.

**Strengths:**

Significance: The proposed Aranyani framework has the potential to be a groundbreaking adjustment in online learning algorithms, paving the way for enhanced fairness in this domain.

Originality: This research stands out due to its unique experimental environment settings and the innovative adjustments made in the problem formulation. The paper presents a fresh perspective by introducing a new idea in the realm of online learning.

Quality: While there are certain ambiguities, primarily concerning the linkage between foundational knowledge and experimental configurations, the paper's narrative is cohesively structured and of commendable quality. The meticulous mathematical proofs provided for each of the Aranyani properties add depth and credibility to the work.

Other Notable Strengths: Aranyani's design, which negates the need to store individual data instances, is especially significant concerning data privacy. This approach not only addresses data storage challenges but also underscores the paper's emphasis on operating based on aggregate statistics rather than individual data points.

**Weaknesses:**

Methodological Clarity:
- The manuscript would be enhanced with a more detailed breakdown of the concluding stages of the training process. It remains uncertain whether the oblique decision tree is designed as an end-to-end neural network or if Aranyani follows the training pattern of standard tree models while adopting a neural configuration.

Performance Insights & Recommendations:
- It would be beneficial to have deeper insights regarding Aranyani's subdued performance on the COMPAS dataset. Could there be challenges in using Aranyani for datasets with less than 7k entries?
- While the paper indicates that Aranyani is more suited to trees of limited height, it would be useful to explore alternative approaches for instances where greater tree height is needed.

Experimental Enhancements:
- Real-World Application: Observing Aranyani's functionality in intricate real-world situations, outside the realm of benchmark datasets, would add depth to the findings.
- To further solidify the paper's comparative analysis, including more recent fairness representation learning models from the related works — modified for online environments — as baselines would be valuable.

Visualization Feedback: In Figure 4 (right), legends are necessary to distinguish which line represents accuracy.

**Questions:**

"In scenarios where the provided dataset is inherently imbalanced, does the definition of fairness shift? Could the fairness criteria evolve based on varying tasks and datasets? And how does the framework handle datasets that may already carry inherent biases? (This query extends broadly to research in the domain of fairness.)"

Regarding addressing some of the weaknesses:
- Handling imbalanced or biased datasets is a nuanced challenge in fairness-related research. Proper preprocessing, data augmentation, or applying techniques like re-sampling can help. If Aranyani uses any such methodologies or has in-built mechanisms to counter these biases, it would be crucial to highlight.
- Furthermore, explaining the adaptability of Aranyani in different scenarios, especially with datasets that are inherently biased, would provide clarity on its real-world applicability.

**Details Of Ethics Concerns:**

I couldn't find any ethical issues with this paper.

---

> ### Author Response · Authors · 2023-11-18
>
> Thank you for your detailed feedback and helpful comments.
>
> > Details of Aranyani’s training procedure.
>
> We have now provided a detailed outline of Aranyani’s training procedure in Appendix C.3 (Algorithm 1).
>
> Yes, Aranyani can be trained as an end-to-end neural network. The task loss gradients can be computed directly using standard autograd libraries. Also, for each input gradient w.r.t. node outputs can be efficiently computed. Then, these node outputs and gradients are aggregated over time to form the final gradients shown in Equation 5. We have implemented Aranyani using TensorFlow.
>
> > Impact of dataset size on Aranyani’s performance
>
> We have performed additional experiments to evaluate the robustness of Aranyani when the dataset size is reduced. We report the performance in Appendix D (Subsection: Dataset Size Ablation). We observe that the tradeoff between fairness and utility gradually deteriorates with a decrease in the dataset size. This happens primarily because it is difficult to achieve higher accuracies with lesser data.
>
> > Impact of tree depth on Aranyani’s performance
>
> In ablation experiments (Section 5.3), we experimented with deep trees up to a height of 8 and investigated its impact on the performance. Figure 4, shows the tradeoff between accuracy (improves with height) and DP (worsens with height) with tree height.
>
> In practice, we have observed that a small tree depth is generally adequate for the majority of tasks. This aligns with findings from prior studies, such as TEL [1] (which employed a tree depth of 4) and DGT [2] (which utilized tree depths up to 10).
>
> [1] Tree Ensemble Layer, Hazimeh et al. 2020.
> [2] Learning Accurate Decision Trees with Bandit Feedback via Quantized Gradient Descent, Karthikeyan et al., 2022.
>
> > The practical utility of Aranyani in real-world applications.
>
> A prominent real-world application for improving fairness in online settings is applications that involve output moderation using safety/toxicity classifiers), which may include chatbots, comments on social media platforms, etc.
>
> This has motivated us to work on this problem entails safety or toxicity classifiers that need to adapt to new safety violation rules/examples on the fly. As such we adapted the toxicity classification task (Civil Comments [3]) to this particular setup and will release the code to reproduce the setup for future work.
>
> We also perform experiments on the CelebA image classification dataset. The task for this dataset is to predict certain characteristics of images (in this case hair color). One can imagine that fairness in predicting image characteristics would be crucial to services that organize user photos. Such image characteristics may be used for organization; fairness ensures consistent organization experience for users.
>
> [3] https://www.kaggle.com/c/jigsaw-unintended-bias-in-toxicity-classification/data
>
> > Comparison of Aranyani with recent batch learning algorithms
>
> Most representation learning methods are trained in an incremental setting where they require access to task identities. Even within a task, these algorithms need access to a batch with samples from different protected groups. Such labeled instances are not available in the online learning scenario.
>
> Due to its stochastic batch optimization procedure, we found that only FERMI [4] can be adapted to function in the online setting. We report the results in Appendix D (Subsection: Comparison with Batch Techniques). We observe that FERMI struggles to perform with batch size=1, and is unable to improve fairness in this setting.
>
> [4] A Stochastic Optimization Framework for Fair Risk Minimization, Lowy et al. 2022
>
>
>
> > Legends in Figure 4 (right)
>
> We have improved the figure and added legends for each metric.

---

> > ### Author Response · Authors · 2023-11-21
> >
> > Thanks for taking the time to review our responses. Please let us know if you have any questions or would like any further clarifications.

---

> > ### Comment · Reviewer_mq1x · 2023-11-21
> > **Impressive Effort in Addressing Comments and Enhancing the Paper**
> >
> > I'm pleasantly surprised that the authors have effectively addressed all the feedback and incorporated additional experimental findings. I believe the paper has significantly improved as a result.

---

> > > ### Author Response · Authors · 2023-11-22
> > >
> > > Thank you very much for reviewing these changes. We greatly appreciate your time and thoughtful feedback on how these changes improve the paper. We also greatly appreciate your support of the paper.

---

### Official Review · Reviewer_vgYA · 2023-11-05

**Soundness:** 3 good
**Presentation:** 4 excellent
**Contribution:** 3 good
**Rating:** 8
**Confidence:** 3

**Summary:**

The authors propose a framework, Aranyani,  based on an ensemble of (soft) oblique decision trees to address group fairness in the online learning setting where one instance arrives at a time. This work proposes a method to update the oblique decision forests model that only relies on updating two aggregated statistics for each node    in the tree and each sensitive group (Section 3.3).  The authors provide a theoretical analysis (Section 4) of the proposed approach such as a bound on demographic parity (DP) based on the depth of the tree under the assumption that the fairness constraint is satisfied on each tree node. Moreover, they also show that the proposed approach to aggregate statistics in an online fashion guarantees gradient norm convergence with enough time steps. Finally, they provide experimental results showing that their approach achieves in general better or competing tradeoffs between accuracy and demographics parity.

**Strengths:**

The paper is well written, the problem is well motivated, and results are promising. I think one of the interesting aspects of the proposed approach, in addition to its online characteristics (no need to store previous data, no need to query the model multiple times), is that the group fairness objective (DP in this case) is being imposed at a local level (in each node). This may reduce the disparities in terms of DP across different regions of the input space. The theoretical analysis provided is reasonable, it motivates and provides some grounded guarantees on the proposed solution.

**Weaknesses:**

The weaknesses I see are summarized in the following questions:

How do you guarantee that in each leaf you have samples from both sensitive groups? From Eq.3 and Eq 5 I understand that $F_{i,j}$ requires examples from both sensitive groups. If this is not guaranteed in the solution, how do you deal with nodes containing samples from a single sensitive group?. How does this method scale with multiple groups?

A discussion on $\delta$ and $\lambda$ parameters from Eq.4 should be provided. In particular, I think $\lambda$ should be set based on $\epsilon$ (DP constraint) from Eq 2, which is the fairness violation that the user is willing to accept. In general, I would have expected to maximize the $\lambda$ parameter until that constrain is satisfied ($\epsilon$). However, it seems that the Huber loss function is not zero when the epsilon constraint is satisfied. Why did you choose this approach to enforce the fairness constraint?


I think the algorithm or a simplified pseudocode of the proposed framework should be provided. This would help with the understanding and reproducibility of the work since it summarizes the logic of the proposed approach.

**Questions:**

In addition to the questions in the weakness section I have some concerns about the practical scalability of this approach.  My understanding is that each time step a new oblique tree is generated, then, the final model can be considerably large. Have the authors thought how this can scale in practice?

---

> ### Author Response · Authors · 2023-11-18
>
> Thank you for your detailed feedback and helpful comments.
>
> > Functioning of Aranyani at each node and its scalability with multiple protected groups
>
> In our work, we use soft decisions at each node of the tree (soft-routed oblique trees defined in Section 2). In this way, every sample has some probability of reaching all leaves. The probability of reaching a node is a function of the outputs in the path from the root to that node. Since we want nodes to assign outputs equally to different groups, we apply fairness constraints on the node outputs.
>
> **Scaling**: The storage cost of the aggregate statistics scales linearly with the number of protected groups. For binary settings, the storage cost is $O(2^hd)$, where $d$ is the data dimension and $h$ is the tree height. When we have $K$ groups, the storage cost becomes $O(2^hKd)$. The time complexity increases by a constant amount when $K$ groups are available, as the Huber loss must be computed for each group (refer to Equation 24 in Appendix B).
>
> > Choice of fairness constraint parameters $\lambda$ and $\delta$
>
> A large number of in-processing fairness regularization techniques apply a divergence with a regularization strength parameter to improve fairness at the cost of model performance (cf. [1, 2]). It is expected that as $\lambda$ grows, the fairness violation will decrease. However, note that in practice, we are interested in test performance vs test fairness violation tradeoffs, and hence tuning the value of $\lambda$ could only be achieved using a validation set as is common in prior work (like FERMI [2]). We follow the same procedure and report results for a range of $\lambda$s.
>
> [1] Fairness-Aware Learning for Continuous Attributes and Treatments, Mary et al. 2019
> [2] A Stochastic Optimization Framework for Fair Risk Minimization, Lowy et al. 2022
>
> > Pseudocode for Aranyani’s training algorithm
>
> We have added a detailed outline of Aranyani’s online algorithm in Appendix C.3 (Algorithm 1).
>
> > Scalability (computational time) of Aranyani
>
> We would like to clarify that we do not generate a new oblique tree at each step. We use a fixed set of complete binary trees (forest) throughout the training process and only the node parameters are updated during training. As is the case for training online models the only updates we use are task loss gradients and fairness loss gradients (Equation 5), which need to be computed at each time step.
>
> In practice, our approach is quite scalable across large data streams. We conducted additional experiments to evaluate the runtime of our algorithm and report them in Table 2 (Appendix D, Subsection: Runtime Analysis).

---

> > ### Author Response · Authors · 2023-11-21
> >
> > Thanks for taking the time to review our responses. Please let us know if you have any questions or would like any further clarifications.

---

### Official Review · Reviewer_9Pu9 · 2023-11-08

**Soundness:** 3 good
**Presentation:** 3 good
**Contribution:** 3 good
**Rating:** 8
**Confidence:** 3

**Summary:**

This paper explores the application of an existing machine learning model to improve group-level fairness performance in an online data streaming setting. Specifically, the authors work in a setting where individual data points arrive at the predictive system one after another over time. Given this setting, the authors deploy a predictive algorithm based on an oblique decision forest. The paper shows how to incorporate group fairness notions such as demographic parity while ensuring efficient training of the forest-based model online. The authors use real-world datasets ranging from tabular to vision and language to highlight the efficacy of their method.

**Strengths:**

The authors propose a novel application of oblique decision trees in the online group fairness setting. I believe the paper has several strengths.

The work is original as it explores the novel application of oblique decision trees in the context of group-fair online predictions. In order to ensure predictions are accurate and fair in the online setting, the authors provide key insights and explore the novel adaptation of oblique decision trees for efficient training. They also introduce theoretical lemmas to help show the relationship of the complexity, fairness performance and the modeling choices, e.g., the depth of the tree.

The authors have presented a work of good quality. The paper provides extensive theoretical complexity analyses. In addition, the authors provide evaluations for demographic parity fairness on multiple modalities of data, e.g., tabular data, image data, and language data. The work also includes extensive ablation studies to empirically show the properties and behavior of the proposed method.

The authors do a good job in making the paper clear to read. The scenario is clearly defined, the intuitions clearly stated, and the modeling is explained in clear language and terms. The article was relatively easy to follow even without understanding of oblique decision trees.

In my opinion, the work can be significant in showing how to adapt a powerful tree-based model for online predictions with fairness as an important consideration. This is especially for tabular data where tree-based models have been shown to perform better. The authors' proposed model may prove to be an useful tool in the practitioner's toolbox for online predictive systems, especially where fairness is also important to consider.

**Weaknesses:**

In terms of weaknesses, I find the following points.
* The evaluations only show demographic parity results. While the authors mention how to formulate Aranyani for other group-fairness notions, they provide no results to highlight how it would perform for notions also based on the true label, e.g., equalized odds or equal opportunity.
* The results do not have error bars. So, it is difficult to compare different methods without taking into consideration the variance across different initialization. As far as I understand, Aranyani and the other baselines may not be guaranteed to find the global optimum. Hence, reporting the mean and the variance across different initialization is very important for fair comparison.
* While the work measures and reports the expectation of the fairness measure at each time step, it is unclear what the fairness dynamic looks like for different methods across time as the training proceeds. In online settings, the variation of the fairness measure over time may tell a story and be essential to see. Specifically, it is not clear how the fairness measure **varies across different time steps**. It is not clear if the training process is relatively stable or not, and how Aranyani compares to the baselines in this respect.
* As per the appendix results, the fairness result variance seems to increase significantly as we move from a single tree to a forest. The authors motivate the need for a forest instead of a tree with a promised reduction of variance. However, in some sense, it seems that variance can increase as well. This aspect does not seem to be discussed well.


There are a few minor points as well.
* Figure 4 (right) includes error bars, but it is not clear how the variance is measured in this case. Is it the variance of a measure across time steps? It is not clear why the tree depth impacting the variance of fairness so greatly.
* The fairness performance seems to degrade considerably when the tree depth is higher. This follows from the theoretical finding, where the fairness bound becomes looser as we increase the tree depth. However, this may limit the expressivity of the model in some situations.
* There are no ablation studies that show the impact of the particular loss that is used to measure fairness, i.e., using Huber loss over $\ell_2$ loss or hinge loss.
* While the authors provide a complexity analysis for their method, an empirical runtime analysis comparing the different methods might have also been insightful.
* Figure 4 (right) does not have any legend. It is difficult to understand which line/symbol corresponds to which metric.

Very minor point.
* The Aranyani triangle symbol in the text does not match with the plots.

**Questions:**

1. The authors showed how to implement a binary oblique decision tree. However, is it possible to formulate a non-binary version of oblique trees for fairness? Are there any scenarios that may necessitate non-binary splits in the tree?
2. The scenario considered exposes the ground-truth label of an individual data point after the prediction is made, irrespective of the predicted label. However, in many real-world online decision-making settings, the ground-truth label is observed **only** when the prediction/decision is positive [1, 2, 3], e.g., in the COMPAS recidivism setting. This is the selective labeling scenario that reflects the real world more. Can the method be applied when true labels are not observed when the prediction is negative?
3. Instead of considering a tree-based structuring across the nodes, can we train a GNN where each node's representation is learned from the features? Would it be possible to apply similar tricks to optimize for the problem?
4. Temporal aspects of fairness can have many different meanings. While the expected fairness across different time steps is one measure, there can be other measures, e.g., fairness in hindsight. Can Aranyani be modified to work for such temporal notions?
5. What does the variance of the fairness look like during the online training across the time steps? Does the fairness measure change a lot from one time step to another? Is it stable, or does it reach a stable state relatively soon? Similarly, what does the accuracy nature look like over time?
6. Why does tree depth impact the fairness variance so much? Similarly, moving from a tree to a forest should reduce variance. But, from Fig. 6, an apparent relationship is not clear. In some cases, it seems that having a forest increases the variance. Do the authors have a thought about this behavior?
5. How would the method compare to bandit-based solutions? Bandits seem well suited to the single-datum online setup discussed here, where the bandit can observe the true label no matter the decision.
6. If we start moving away from the one datum at a time setup and consider multiple data points arriving at each time step, would the efficacy of the oblique tree reduce compared to other methods, e.g., MLPs?

[1] Kilbertus, Niki, et al. "Fair decisions despite imperfect predictions." International Conference on Artificial Intelligence and Statistics. PMLR, 2020.

[2] Rateike, Miriam, et al. "Don’t throw it away! the utility of unlabeled data in fair decision making." Proceedings of the 2022 ACM Conference on Fairness, Accountability, and Transparency. 2022.

[3] Wick, Michael, and Jean-Baptiste Tristan. "Unlocking fairness: a trade-off revisited." Advances in neural information processing systems 32 (2019).

---

> ### Author Response · Authors · 2023-11-18
>
> Thank you for such detailed feedback and so many insightful suggestions.
>
>
> > Efficacy of Aranyani using other notions of fairness
>
> We have added the results for the equality of opportunity fairness measure in Appendix D (Subsection: Equality of Opportunity). Even in this setup, we find that Aranyani achieves a strong accuracy-EO tradeoff and outperforms the majority baseline significantly.
>
>
> > Error bars in Aranyani’s results
>
> We have incorporated the variance for each result and revised the figures in both Figures 2 and 3 within the main paper.
>
> > Variance in accuracy and DP scores during the online training process.
>
> We have performed additional experiments to examine this scenario in Appendix D (Subsection: Performance Variance). We report the results from 5 different runs on the Adult dataset. We observe that the variation in the metrics: accuracy and DP, exist only during the initial phase of the online learning process (please note that the $x$-axis is in log-scale for better visibility). After as few as 1000 samples, both the accuracy and DP scores stabilize with minimal variation.
>
> > Variance in results with an increase in the number of trees in the forest.
>
> Thank you for pointing this out. In the initial draft, we reported the minimum and maximum scores out of 3 runs by the error bars. We have fixed that now and report the standard deviation across 5 different runs for each tree count configuration. We have also expanded our evaluation to investigate up to 15 trees in the forest. In the updated figure, we observe a decreasing trend in the standard deviation of the reported metrics with an increase in tree count, as hypothesized in the main paper (Section 2).
>
> > The influence of tree depth on the expressivity of Aranyani.
>
> The empirical results for accuracy and fairness follow that of our theoretical results. Apart from the tree depth, there are several ways to increase the expressivity of the model: (a) using more trees, and (b) using stronger classifiers at the node level, among others. In practice, we have observed that a small tree depth is generally adequate for the majority of tasks. This aligns with findings from prior studies, such as TEL [1] (which employed a tree depth of 4) and DGT [2] (which utilized tree depths up to 10).
>
> [1] Tree Ensemble Layer, Hazimeh et al. 2020.
> [2] Learning Accurate Decision Trees with Bandit Feedback via Quantized Gradient Descent, Karthikeyan et al., 2022.
>
> > Ablation with different regularization loss functions.
>
> In this work, we focus on the Huber loss as the regularization measure. We choose Huber as it provides us with better theoretical guarantees while using stale gradients. In general, it is possible to explore other divergence measures like L2 loss.  If we use L2 loss instead the fairness gradient in Equation 5 will have the $F_{ij} \Delta F_{ij}$ term only. The approximation error of $F_{ij}$ may increase over time leading to a worse convergence bound of the gradients. For Huber loss, it is possible to theoretically bound the error using the $\delta$ parameter.  Hinge loss on the other hand is used to measure the classification performance and we could not think of how to apply it directly to regularize $F_{ij}$.
>
> We have also added the results and discussion for this experiment in Appendix D (Subsection: Regularizer Ablation).
>
> > Empirical runtime analysis of Aranyani
>
> We have provided an empirical estimate for the runtime in Appendix D (Runtime analysis). Please refer to that section for a detailed description.
>
> > Legends in Figure 4 (right)
>
> We have updated the figure and included legends to improve its clarity.
>
> > Typo in Aranyani’s symbol
>
> We have fixed the symbol in the current draft.
>
> > Scaling Aranyani to an M-ary tree and its applications
>
> Yes, it is possible to have a non-binary, $M$-ary oblique tree structure. For an $M$-ary oblique tree, the DP bound is $hM^h\epsilon$ and the empirical Rademacher complexity bound is $M^h/\sqrt{n}$. Again, this goes on to show the tradeoff between fairness and accuracy as the DP bound worsens and the Rademacher bound improves.
>
> We do not foresee any specific scenario that necessitates the use of $M$-ary splits. Although $M$-ary splits lead to better expressivity of the model, it can be achieved through various means like increasing the tree depth, number of trees in the forest, complexity of the node networks, etc.
>
> > Ablations with different feedback mechanisms during online learning
>
> We have performed this experiment on the COMPAS dataset, where the model receives gradient updates only when its prediction $\hat{y}=1$. We report the results in Appendix D (Limited Feedback) section. We observe that Aranyani achieves a similar tradeoff curve in this setup. However, it is unable to achieve higher accuracies as it receives a significantly smaller number of gradient updates.

---

> > ### Author Response · Authors · 2023-11-18
> >
> > > Expanding Aranyani's tree structure to a graph structure.
> >
> > Thank you for this suggestion. Generalizing from a tree structure to a graphical structure is an interesting direction. However, it is not obvious to us what the fairness constraints in the intermediate node levels should be. In the tree structure, the node outputs direct an input instance to one of the paths and it is possible to apply fairness constraints on them. The MLP variant of Aranyani can be considered a step towards the graph structure where we use a fully connected MLP but each parameter receives a fairness gradient update based on the final network inputs. Future works can explore this extension to graphical structures and its implications on fairness-utility tradeoffs.
> >
> > > Temporal analysis in the performance of Aranyani.
> >
> > Fairness in Hindsight [3] was originally defined for the notion of individual fairness, where we wish to understand how the treatment of an individual varies over time. At this point, it is not entirely clear to us how to extend this to the group fairness notion in online learning. This is primarily because of two reasons: (a) group fairness isn’t defined at the level of individual inputs, and (b) samples of each group change continuously during online learning.
> >
> > As an initial step towards this notion, we maintain a held-out set of test samples and observe the fairness loss on this set over time. This held-out set never appears in the online stream. We have performed this experiment and report the results in Appendix D (Subsection: Temporal Analysis).
> >
> > [3] Individual Fairness in Hindsight, Gupta et al, 2022
> >
> > > Variation in Aranyani’s performance during online training
> >
> > We have performed additional experiments to examine this scenario in Appendix D (Subsection: Performance Variance). We have also included a discussion of the results.
> >
> > > The influence of tree depth on the variability in Aranyani's performance.
> >
> > We have updated the results in Figure 4 (right) to include the standard deviation computed across 5 runs (with different random initialization). We observe that the variance is high when the tree depth is very small, depth=2. Here the complexity of the model is low, and it is neither overfitting nor underfitting showcasing lower accuracy (high bias) and high variance. As the depth increases, the model starts to converge better and the variance decreases. At much higher depths, the variance increases again showcasing that the model may have overfit the data.
> >
> > > Comparison with bandit-based solutions.
> >
> > Bandit-based systems function by updating their decision-making system based on the rewards received from the environment. For group fairness, it is not clear how to define the reward for a single input instance. This is because group fairness is defined across groups of samples. Bandit-based solutions can function in the batch setting but currently, it is unclear how to extend it to the online setting.
> >
> > > Performance of Aranyani in batch settings.
> >
> > We have performed experiments in the batch setting on the Adult dataset with batch size=10. We report the results in Appendix D (Batch Learning). We observe that Aranyani performs significantly better than other approaches even in this setup.
> >
> > We would like to thank you again for your thorough and detailed review.

---

> > > ### Author Response · Authors · 2023-11-21
> > >
> > > Thanks for taking the time to review our responses. Please let us know if you have any questions or would like any further clarifications.

---

> > > > ### Comment · Reviewer_9Pu9 · 2023-11-21
> > > >
> > > > I thank the authors for answering all of my comments and concerns. I especially appreciate the authors' effort to perform additional experiments to address my comments and other reviewers' comments. I believe that the changes and additional results significantly improve the clarity and quality of the paper. The error bars in Figures 2 and 3 helped me understand and compare Aranyani to the other methods. The variance in Figures 2 and 3 also aligns with my intuitions, i.e., the variance in COMPAS and CelebA are significantly higher because the ground truth and sensitive class labels are highly skewed in these datasets. I think this opens an exciting avenue to explore reducing this variance in the realm of fair online learning. Similarly, Figure 8 in the Appendix has helped me better understand Aranyani's learning progress and what the temporal variance looks like during the learning process. Similarly, the performance under adversarial data streams where the data distribution shifts improves the transparency regarding the limitations of the work and the potential for further exploration.
> > > >
> > > > Owing to the changes, I am willing to increase my score for the paper.

---

> > > > > ### Author Response · Authors · 2023-11-22
> > > > >
> > > > > Thank you very much for reviewing these changes. We greatly appreciate your time and thoughtful feedback on how these changes improve the paper. We also greatly appreciate your support of the paper and increase in review score.

---

### Author Response · Authors · 2023-11-18
**General Response**

We would like to thank the reviewers for taking the time to go through our paper and provide detailed feedback/suggestions. We have conducted additional experiments and revised the paper to address any concerns. Here is the summary of all the changes:

1. We have included the error bars for Aranyani’s results in Figure 2 and Figure 3.
2. We have extended the ablation experiments to investigate the impact of the number of trees in the forest on performance variance (Appendix D, Subsection: Tree Ablations, Figure 6).
3. Experiments to investigate Aranyani’s performance variance during the online learning process (Appendix D, Subsection: Performance Variance, Figure 8).
4. We report Aranyani’s performance under an adversarial input stream (Appendix D, Subsection: Adversarial Stream, Figure 9).
5. We report Aranyani’s performance under limited feedback/gradient updates (Appendix D, Subsection: Limited Feedback, Figure 10 (left)).
6. We report Aranyani’s performance with a different fairness notion: Equality of Opportunity (Appendix D, Subsection: Equality of Opportunity, Figure 10 (center)).
7. We report Aranyani’s performance in the batch learning setting (Appendix D, Subsection: Batch Learning, Figure 10 (right)).
8. We report the runtime of Aranyani and other baselines (Appendix D, Subsection: Runtime Analysis, Table 2).
9. Ablation experiment with different fairness regularization loss (Appendix D, Subsection: Regularizer Ablation, Figure 11 (left)).
10. Ablation experiment with different dataset sizes (Appendix D, Subsection: Dataset Size Ablation, Figure 11 (center)).
11. Comparison with offline batch learning techniques (Appendix D, Subsection: Comparison with batch learning, Figure 11 (right)).
12. We report Aranyani’s performance on a held-out set during online learning (Appendix D, Subsection: Temporal Analysis, Figure 12 (left & center)).
13. We investigate the convergence of Aranyani’s tradeoff curve (Appendix D, Subsection: Tradeoff Convergence, Figure 12 (right)).
14. We have included an algorithm to outline Aranyani’s training procedure (Appendix C.3, Algorithm 1).
15. We have included the results for the FAHT baseline for all datasets in Figure 2.
16. We have updated the results and their presentation in Figure 4 (right). The reported results now present the variance across 5 different runs.

---

### Meta-Review · Area_Chair_Nxnb · 2023-12-10

**Metareview:**

The paper addresses the the problem of fairness in online decision-making, which they solve with an algorithm based on an ensemble of decision-forest that provide state-of-the-art results. The reviewers (and I concur) highlight the novelty, clarity, theoretical analysis and empirical results of the paper. Importantly, while the reviewers raised several points of concerns during their initial review, the authors have done a good job at addressing the major points raised, which they used to improve their paper. Thus, I recommend acceptance.

**Justification For Why Not Higher Score:**

I believe that this is a complete paper (even more now with the additional details in the appendix) on an important and still understudied paper. However, the impact within the ICLRT community may still be limited and this is why I believe that spotlight would be the right level of visibility for this paper.

**Justification For Why Not Lower Score:**

I believe that the paper is a clear acceptance case, being the points raised by the most negative reviewer, in my opinion not major (or even justified). Also, I believe that this is a complete paper (even more now with the additional details in the appendix) on an important and still understudied paper. I believe that the good job made by the authors (on both the paper and the rebuttal period) deserves a spotlight.

---

### Decision · Program_Chairs · 2024-01-16

Accept (spotlight)